# Loss of BCL9/9l suppresses Wnt driven tumourigenesis in models that recapitulate human cancer

David M. Gay[1], Rachel A. Ridgway[1], Miryam Müller[1], Michael C. Hodder[1], Ann Hedley[1], William Clark[1], Joshua D. Leach[1], Rene Jackstadt[1], Colin Nixon[1], David J. Huels[1,4], Andrew D. Campbell[1], Thomas G. Bird [1,2,3] & Owen J. Sansom [1,2]

Different thresholds of Wnt signalling are thought to drive stem cell maintenance, regeneration, differentiation and cancer. However, the principle that oncogenic Wnt signalling could be specifically targeted remains controversial. Here we examine the requirement of BCL9/9l, constituents of the Wnt-enhanceosome, for intestinal transformation following loss of the tumour suppressor APC. Although required for Lgr5+ intestinal stem cells and regeneration, Bcl9/9l deletion has no impact upon normal intestinal homeostasis. Loss of BCL9/9l suppressed many features of acute APC loss and subsequent Wnt pathway deregulation in vivo. This resulted in a level of Wnt pathway activation that favoured tumour initiation in the proximal small intestine (SI) and blocked tumour growth in the colon. Furthermore, Bcl9/9l deletion completely abrogated β-catenin driven intestinal and hepatocellular transformation. We speculate these results support the *just-right* hypothesis of Wnt–driven tumour formation. Importantly, loss of BCL9/9l is particularly effective at blocking colonic tumourigenesis and mutations that most resemble those that occur in human cancer.

---

[1] Cancer Research UK Beatson Institute, Garscube Estate, Switchback Road, Glasgow G61 1BD, UK. [2] Institute of Cancer Sciences, University of Glasgow, Garscube Estate, Glasgow G61 1QH, UK. [3] MRC Centre for Inflammation Research, The Queen's Medical Research Institute, University of Edinburgh, Edinburgh EH16 4TJ, UK. [4] Present address: Academic Medical Center (AMC), University of Amsterdam, Amsterdam 1105 AZ, The Netherlands. Correspondence and requests for materials should be addressed to O.J.S. (email: o.sansom@beatson.gla.ac.uk)

Deregulated Wnt signalling is a hallmark of colorectal cancer (CRC). This predominantly results from mutations in the tumour suppressor gene adenomatous polyposis coli (*APC*), which is found in 80% of the patients[1]. APC is a negative regulator of the canonical Wnt signalling pathway, forming part of the β-catenin destruction complex, with frequent mutation in CRC resulting in the hyperactivation of the pathway[2,3]. In the absence of Wnt signalling, APC associates with AXIN, Casein Kinase 1 (CK1) and glycogen synthase kinase 3 beta (GSK3β), which are required for the phosphorylation of β-catenin—marking it for ubiquitination and degradation[4]. Following Wnt activation or *APC* mutation, the complex is inactivated, whereby phosphorylated β-catenin can no longer be ubiquitinated, saturates the destruction complex and allows de novo synthesised β-catenin to translocate to the nucleus[5]. Nuclear β-catenin interacts with T-cell factor-1/lymphoid enhancer factor-1 (TCF/LEF1) transcription factors to drive target gene expression[6,7]. Additional transcriptional co-activators of β-catenin such as B-cell lymphoma 9 (BCL9)[8] and Pygopus[9] co-operate in β-catenin-mediated transcription, forming part of the Wnt enhanceosome[10].

The majority of *APC* mutations cluster in a specific region of the 5′ end of the gene, known as the mutation cluster region (MCR)[11]. The MCR encodes the 20 amino acid repeats (20AARs) which are required for β-catenin binding and degradation[12] and are truncated in CRC, leading to hyperactivated Wnt signalling. Interestingly, colon tumours retain on average two 20AARs[13], thought to result in a 'just-right' level of Wnt signalling, which may be sub-maximal[14]. There is evidence that the number of retained 20AARs influences CRC tumour location: proximal colonic tumours retained more than distal colonic tumours[15,16]. This tumour distribution could be influenced by the decreasing Wnt gradient that runs from the proximal to distal colon[15]. Leedham and colleagues proposed where tumours have high pathological Wnt signalling, proximal colonic tumour formation is unfavourable due to high underlying basal Wnt signalling levels in that region, instead distal colonic tumorigenesis is favoured[15]. Moreover, we recently showed that pharmacological reduction of Wnt signalling reduced intestinal stem cell (ISC) number, ISC competition and increased proximal small intestinal tumour formation in mice where *Apc* was deleted in the ISCs[17] These studies suggest that colon tumours select for *APC* mutations providing the optimal level of Wnt signalling and that Wnt signalling influences the size of the ISC pool as well as ISC competition.

There has been limited success in targeting Wnt signalling in CRC. Whilst some Wnt-driven cancers, such as those with *RNF43* mutations or *RSPO* amplifications, appear sensitive to suppression of extracellular Wnt signalling using LRP6 blocking antibodies or Porcupine inhibition[18,19], these mutations are rare in CRC. Importantly, as the majority of CRCs carry *APC* mutations and are Wnt-ligand independent, there is a need to develop strategies that inhibit Wnt signalling in a ligand-independent manner[20]. This said, Tankyrase inhibitors, which stabilise AXIN, while exhibiting efficacy in CRC cell lines, have severe intestinal toxicity in vivo[21,22]. Additionally, cells that experience chronic Wnt signalling, including *APC*-mutant CRC cells are refractory to Tankyrase inhibition, due to the significant expression of BCL9l and LEF1 which shield β-catenin from AXIN-mediated destruction[23]. These studies highlight that successful Wnt-based therapies in CRC must act downstream of the destruction complex; disrupting binding of β-catenin to transcriptional activators. Importantly, proof of concept studies has shown that restoration of APC in aggressive carcinoma of mice causes tumour regression[24].

The β-catenin–TCF interface is large and dynamic, making it difficult to target[25]. However, significant interest exists in targeting the β-catenin–BCL9 association in the Wnt enhanceosome. BCL9 and BCL9l, functionally redundant mammalian homologues of the *Drosophila* gene *legless*, play a role in nuclear shuttling of β-catenin and promotion of β-catenin-dependent transcription[8,26,27]. While constitutive deletion of *Bcl9* and *Bcl9l* is embryonically lethal[28], conditional deletion in the murine intestine is tolerated[29]. Deletion of *Bcl9* and *Bcl9l* reduces colonic regeneration following acute colitis and decreases expression of Wnt target genes and ISC markers in colonic tumours generated by chemical carcinogenesis[29]. Hence, BCL9 and BCL9l have been proposed to regulate stemness within the intestinal crypts[30]. Furthermore, both are upregulated in human CRC[31,32] and overexpression of BCL9l significantly increased tumour formation in *Apc*[Min/+] mice[33]. A number of small molecules targeting the β-catenin BCL9 interface have shown promise in APC-deficient cells both in vitro and in vivo[23,34], revealing that targeting of BCL9 and BCL9l may offer a therapeutic window in CRC.

As previous studies investigating BCL9 and BCL9l (BCL9/9l) were performed in colitis-associated cancer models, we wished to investigate the effect in models of cancer directly driven by activation of Wnt signalling either by *Apc* gene deletion or β-catenin stabilisation. We also sought to identify differences in the activation of oncogenic Wnt signalling when compared to homeostatic Wnt signalling to determine whether there was a therapeutic window for Wnt pathway inhibition following a mutation in the pathway. We report that deletion of *Bcl9/9l* sensitises the murine epithelium to perturbation of the Wnt pathway and impacts the Lgr5-ISC population. We show that BCL9/9l are required for the acute transformation of the intestine following homozygous deletion of *Apc* and for Wnt-driven transcriptional programmes associated with APC loss. Unexpectedly, we found that deletion of *Bcl9/9l* accelerated an APC-driven model of intestinal tumorigenesis and favoured adenoma formation within the proximal SI, but suppressed colonic tumour growth. However, if the β-catenin destruction complex is intact, BCL9/9l are absolutely required for mutant β-catenin-driven intestinal and hepatic transformation driven by mutant β-catenin. Moreover, Mieszczanek et al. (co-submitted manuscript) show that if mice carry a truncating mutation in *Apc* that is equivalent to human CRC, loss of *Bcl9/9l* makes these mice resistant to tumorigenesis. Crucially, we show that it is possible to reduce Wnt signalling to a level which prevents transformation in cancer cells which carry ligand-independent Wnt activating mutations, without disrupting normal homeostasis.

## Results

**BCL9/9l control intestinal Lgr5 expression**. The ISC pool is regulated by Wnt signalling[17,35], with the highest levels defining the number of Lgr5 positive ISCs. Given that BCL9/9l are implicated in Wnt signalling and are required for colonic regeneration following acute colitis, we sought to determine whether Wnt signalling is perturbed in BCL9/9l-deficient intestines. To do this we generated *VillinCre*[ER] *Bcl9*[fl/fl] *Bcl9l*[fl/fl] mice. Mice were injected with tamoxifen to induce intestinal Cre-mediated recombination, and harvested 4 days post-induction. Immunohistochemical staining confirmed accumulation of nuclear β-catenin at the base of small intestinal crypts following deletion of *Bcl9/9l* (Fig. 1a and Supplementary Figure 1a). Since BCL9/9l have been implicated in the nuclear shuttling of β-catenin, we performed a proximity ligation assay for E-cadherin: β-catenin complexes at the membrane. We observed that deletion of *Bcl9/9l* results in a significant increase in the number of

E-cadherin: β-catenin complexes in small intestinal crypts compared with WT controls (Fig. 1b, c). Importantly, there was no difference in the expression of *Ctnnb1* and *Cdh1* between WT and BCL9/l deficient crypts (Fig. 1d). These data suggest that there is a reduction in nuclear β-catenin following deletion of *Bcl9/9l*. To investigate whether this correlated with reduced Wnt signalling, we performed RNAseq on intestinal tissue from WT and *VillinCre*[ER] *Bcl9*[fl/fl] *Bcl9l*[fl/fl] mice. Gene Set Enrichment Analysis (GSEA) revealed a negative enrichment for the Lgr5+ ISC gene signature (Fig. 1e and Supplementary Table 1). We confirmed reduced *Lgr5* expression in BCL9/l deficient crypts, while other Wnt target genes including *Cd44*, *Axin2* and *c-Myc* or stem cell markers, *Olfm4*, were not altered (Fig. 1f; Supplementary Figure 1a & 1b). These data are consistent with our recent study and the work by the Kuo laboratory which showed that the Lgr5+ ISCs are the most dependent on the highest levels of Wnt signalling in the intestine[17,35]. This suggested that there may exist a therapeutic window for BCL9/9l loss in the intestine, with deletion preferentially impacting those cells most dependent upon higher levels of Wnt signalling, such as during regeneration or transformation.

**BCL9/9l contribute to intestinal stem cell fitness**. It has been shown that ISCs exhibit neutral drift dynamics[36,37] and that replacement of one stem cell by a neighbour is a stochastic process. Clonal fitness of ISCs has been investigated by tracing the clonal expansion of stem cells over time[38]. To determine whether the loss of the Lgr5 gene signature following deletion of BCL9/9l is functionally relevant in terms of ISC fitness, we induced Cre recombination in *Lgr5-EGFP-Cre*[ER] *tdTom*[fl/+] and *Lgr5-EGFP-Cre*[ER] *tdTom*[fl/+] *Bcl9*[fl/fl] *Bcl9l*[fl/fl] mice with 0.15 mg tamoxifen, so as to recombine, on average, in a single ISC per crypt, labelling that ISC with a tomato reporter (Fig. 2a). We observed a significant reduction in average clone size at 4 and 21 days post-induction following deletion of *Bcl9/9l* compared with WT ISCs (Fig. 2b). Moreover, we observed a profound increase in the number of partially fixed crypts at 21 days post-induction arising from BCL9/9l-null ISCs compared with WT ISCs (Fig. 2c). Interestingly, of fully fixed clones (completely tomato-positive crypts) from *Lgr5-EGFP-Cre*[ER] *tdTom*[fl/+] *Bcl9*[fl/fl] *Bcl9l*[fl/fl] mice, 31 out of 35 retained *Lgr5* expression, indicating that these crypts may have escaped recombination (Fig. 2d). Therefore, the apparent reduction in ISC fitness observed in BCL9/9l-deficient ISCs may be an underestimate of the true effect. Moreover, at day 21 there is a significant reduction in the number of tomato positive crypts in *Lgr5-EGFP-Cre*[ER] *tdTom*[fl/+] *Bcl9*[fl/fl] *Bcl9l*[fl/fl] mice compared with controls, suggesting that BCL9/9l-deficient ISCs have been replaced by WT neighbours (Fig. 2e). These data suggest that BCL9/9l are required for the efficient function of Lgr5-positive stem cells, and that in their absence, ISCs have reduced fitness compared with WT ISCs. Importantly, despite this ISC phenotype, we showed that deletion of *Bcl9/9l* does not perturb intestinal homeostasis (Supplementary Figure 2a & b) in agreement with other studies[29].

**BCL9/9l are required for intestinal regeneration**. We next addressed whether the loss of BCL9/9l could affect other phenotypes associated with Wnt signalling in the intestine. To determine whether *Bcl9/9l* deletion sensitises the SI to a further reduction of Wnt signalling, we treated *VillinCre*[ER] *Bcl9*[fl/fl] *Bcl9l*[fl/fl] mice with the porcupine inhibitor LGK974, blocking Wnt ligand secretion[19]. This resulted in severe crypt atrophy and loss of proliferation after just 3 days (Fig. 3a, b; Supplementary Figure 3c). Vehicle-treated mice displayed no phenotype even after 30 days of treatment (Fig. 3a, b), with previous experiments

demonstrating that LGK974 treatment is well tolerated in wild-type mice up to 50 days and beyond[17]. Wnt signalling is also essential for crypt/organoid culture in vitro. Importantly, BCL9/9l-deficient small intestinal crypts fail to establish in vitro (Supplementary Figure 3a & b).

Finally, the ability of the intestine to regenerate after insult is also Wnt-dependent. Therefore, we investigated whether BCL9/9l are required for intestinal regeneration following irradiation. To this end, Cre-induced *VillinCre*[ER] *Bcl9*[fl/fl] *Bcl9l*[fl/fl] and wildtype mice were culled at a time-point 72 h post-irradiation (10 Gy γ-radiation). Histological analysis revealed that there was a significant reduction in the number of regenerating small intestinal crypts following deletion of *Bcl9/9l* (Fig. 3c, d), indicating their requirement for intestinal regeneration. Together, these data suggest that whilst dispensable for intestinal homeostasis, BCL9/9l are absolutely required following deregulation of the Wnt pathway in the intestine.

**BCL9/9l are required for acute intestinal transformation**. We next investigated the role for BCL9/9l in intestinal neoplasia, a process characterised by hyperactivated Wnt signalling following APC loss.

Homozygous deletion of *Apc* throughout the murine intestine leads to the acute transformation of the epithelium. The ensuing *crypt-progenitor phenotype* consists of significantly increased proliferation, perturbed differentiation and migration, and is maximally penetrant at 4 days post Cre-induced *Apc* deletion[39]. Therefore, we generated *VillinCre*[ER] *Apc*[fl/fl] *Bcl9*[fl/fl] *Bcl9l*[fl/fl] animals. Coincident deletion of *Bcl9/9l* with APC loss strongly suppressed the *crypt-progenitor phenotype*, with a significant reduction in proliferation in both the SI and colon compared to controls (Fig. 4a–c and Supplementary Figure 4a–c). To determine whether this reduction in proliferation was due to a larger reduction in Wnt target gene expression, we performed RNAseq on intestinal tissue from Cre-induced *VillinCre*[ER] *Apc*[fl/fl] and *VillinCre*[ER] *Apc*[fl/fl] *Bcl9*[fl/fl] *Bcl9l*[fl/fl] mice. This revealed downregulation of a number of Wnt target genes following *Bcl9/9l* deletion. Subsequent GSEA confirmed negative enrichment of genes upregulated following acute *Apc* deletion (Fig. 4d and Supplementary Table 2). These downregulated Wnt target genes including *Lgr5*, *Axin2*, *Cd44* and SOX9, were confirmed through RNA in situ hybridisation (RNAscope), immunohistochemical staining and qPCR (Fig. 4e–g; Supplementary Figure 4e). Importantly, many of these genes were unaffected in the normal intestine following *Bcl9/9l* deletion (e.g. *Axin2*, *Cd44* and *Sox9*). Deletion of *Apc* disrupts the destruction complex, stabilising β-catenin, allowing translocation to the nucleus. Immunohistochemical staining confirmed accumulation of nuclear β-catenin within the small intestinal crypts of both Cre-induced *VillinCre*[ER] *Apc*[fl/fl] and *VillinCre*[ER] *Apc*[fl/fl] *Bcl9*[fl/fl] *Bcl9l*[fl/fl] mice (Supplementary Figure 4d). This suggests that BCL9/9l play a critical role in the nucleus as part of the Wnt enhanceosome, alongside their role in shuttling β-catenin. Interestingly, RNAseq data from WT vs *VillinCre*[ER] *Bcl9*[fl/fl] *Bcl9l*[fl/fl] and *VillinCre*[ER] *Apc*[fl/fl] vs Cre-induced *VillinCre*[ER] *Apc*[fl/fl] *Bcl9*[fl/fl] *Bcl9l*[fl/fl] mice revealed that BCL9/9l regulate many more genes following APC loss, with 129 genes differentially expressed between WT and BCL9/9l deficient intestinal epithelia and 655 differentially expressed genes between intestines from Cre-induced *VillinCre*[ER] *Apc*[fl/fl] and *VillinCre*[ER] *Apc*[fl/fl] *Bcl9*[fl/fl] *Bcl9l*[fl/fl] mice (Supplementary Figure 4f). Indeed, only 57 differentially expressed genes are shared between the two datasets (Supplementary Figure 4f). Hence, in homeostasis the main role of BCL9/9l appears to be control of the Lgr5-positive ISC pool, while upon transformation of the epithelium following APC loss, BCL9/9l are required to drive oncogenic Wnt

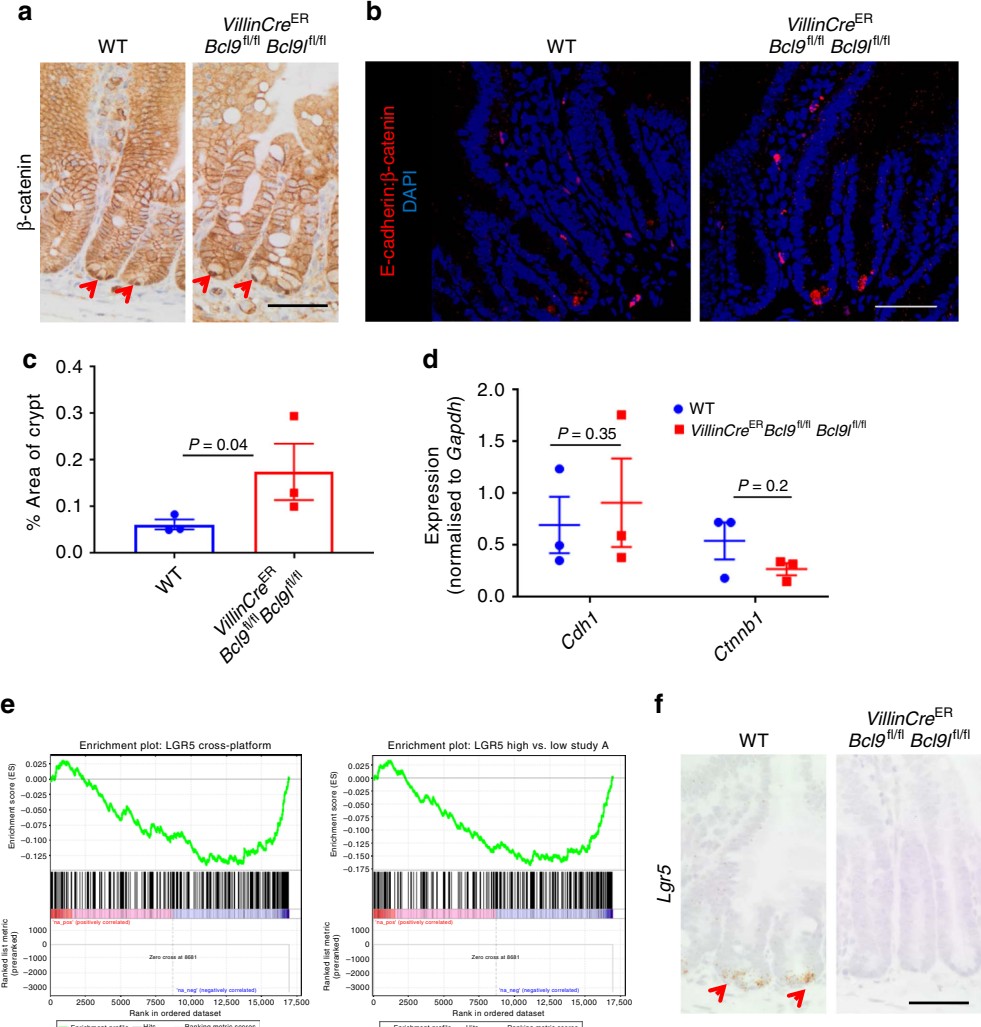

**Fig. 1** BCL9/9l control intestinal Lgr5 expression. **a** Representative β-catenin staining of small intestinal sections from Cre-induced WT and *VillinCre*ER *Bcl9*fl/fl *Bcl9l*fl/fl mice sampled 4 days post tamoxifen injection. Red arrows indicate nuclear β-catenin at the base of the crypt. Scale bar = 50 μm. **b** Proximity ligation assay for E-cadherin: β-catenin complexes (red) from small intestinal sections of small intestinal sections from mice described in (**a**). Nuclei stained with DAPI (blue). Paneth cells appear red, but were excluded from analysis. Scale bar = 50 μm. **c** Quantification of proximity ligation assay, % positive area per crypt (E-cadherin: β-catenin) was quantified and 10 crypts were scored per mouse, *n* = 3 per group, one-way Mann–Whitney *U* test, *P* = 0.04. Data displayed as mean ±SEM. **d** qPCR for *Cdh1* and *Ctnnb1* expression in small intestine of mice described in (**a**), *n* = 3 per group, one-way Mann–Whitney *U* test, *P* = 0.35 (*Cdh1*) and *P* = 0.2 (*Ctnnb1*). Data displayed as mean ±SEM. **e** Gene Set Enrichment Analysis of RNAseq data obtained from small intestinal tissue from WT and *VillinCre*ER *Bcl9*fl/fl *Bcl9l*fl/fl mice, *n* = 3 per group. **f** Representative *Lgr5*-RNAscope and β-catenin staining of small intestinal sections from mice described in (**a**). Red arrows indicate *Lgr5*-staining at the base of the crypt. Scale bar = 50 μm

transcriptional programmes and consequently acquisition of the *crypt-progenitor phenotype*.

**Deletion of *Bcl9/9l* alters intestinal tumour distribution.** We demonstrated that inhibition of Porcupine resulted in reduced ISC number and a rapid fixation of mutant clones[17]. This accelerated tumorigenesis following APC loss due to rapid clonal fixation of crypts, and increased proximal small intestinal lesion number. Given that *Bcl9/9l* deletion caused a very similar phenotype, reduction of the Lgr5-positive ISC pool, one might predict they would phenocopy this data. However, since *Bcl9/9l* deletion was associated with a reduction of Wnt target gene expression following APC loss and a suppression of the *crypt-progenitor phenotype*, one might also predict reduced tumorigenesis. We examined whether loss of BCL9/9l modified tumorigenesis in *VillinCre*ER *Apc*fl/+ mice. Deletion of *Bcl9/9l* significantly accelerated intestinal tumorigenesis and reduced survival (Fig. 5a). This reduction in survival was due to

a significant increase in tumour burden—with mice developing hundreds of small lesions (Fig. 5b–d). Moreover, deletion of *Bcl9/9l* resulted in a profound alteration in tumour distribution. Tumours were uniformly distributed between the proximal and distal SI in Cre-induced *VillinCre*ER *Apc*fl/+ mice, while upon deletion of *Bcl9/9l* there was a significant increase in the number of proximal intestinal tumours (Fig. 5e). The tumours that arose were deficient for BCL9, and despite being positive for nuclear β-catenin, were also negative for *Lgr5* (Fig. 5f). Next, we sampled Cre-induced *VillinCre*ER *Apc*fl/+ and *VillinCre*ER *Apc*fl/+ *Bcl9*fl/fl *Bcl9l*fl/fl mice at 50 days post-induction to confirm an increased rate of tumour formation. The number of both macroscopic and microscopic lesions was significantly increased in BCL9/9l-null intestines, with a concomitant shift towards proximal small intestinal lesion formation also observed (Supplementary Figure 5a & b).

To determine a role for BCL9/9l in colonic tumour growth we injected 4-hydroxy tamoxifen into the colonic submucosa of *VillinCre*ER *Apc*fl/fl and *VillinCre*ER *Apc*fl/fl *Bcl9*fl/fl *Bcl9l*fl/fl mice to

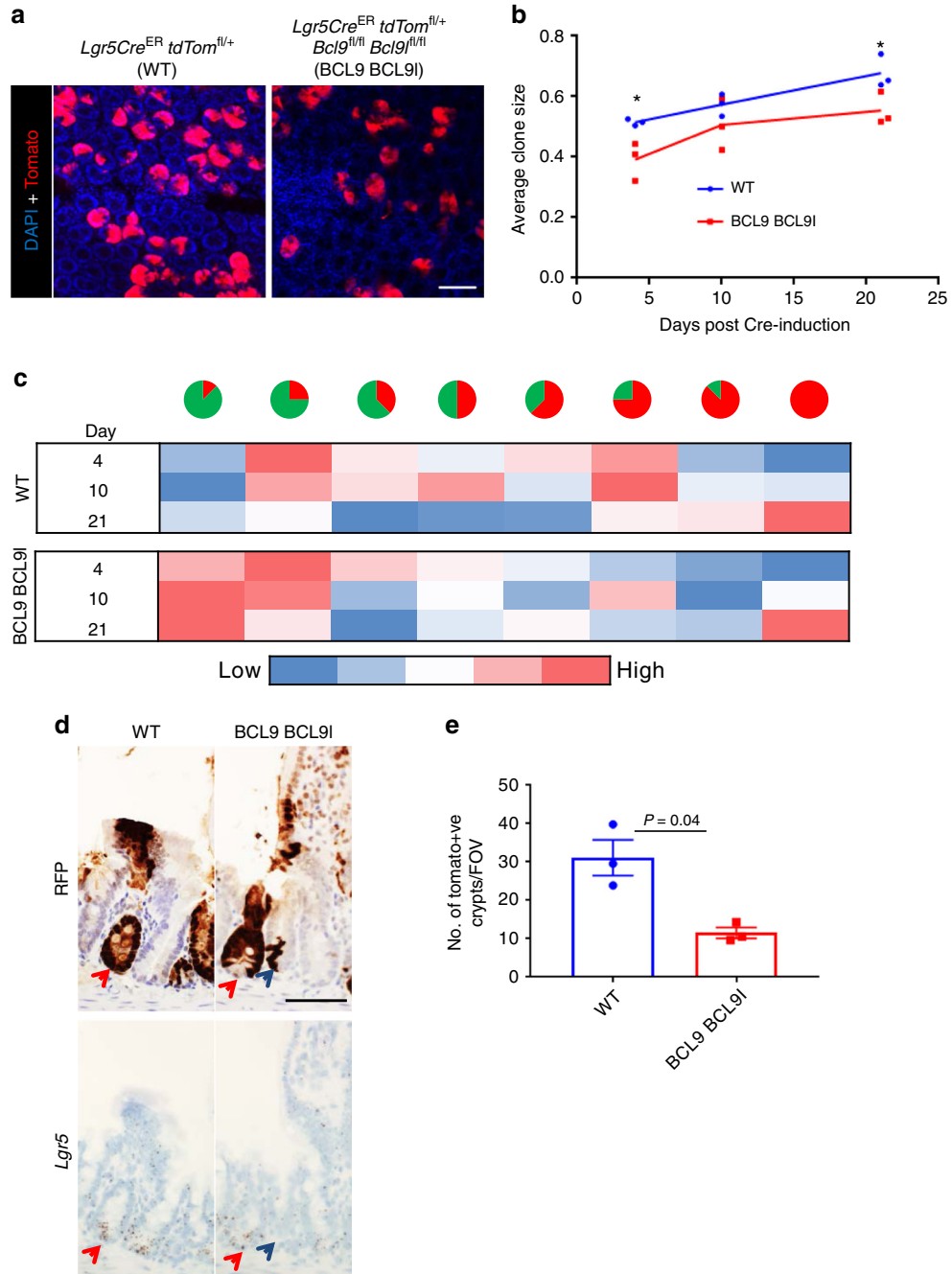

**Fig. 2** BCL9/9l contribute to intestinal stem cell fitness. **a** Representative images from *Lgr5-EGFP-Cre*ER *tdTom*fl/+ (WT) and *Lgr5-EGFP-Cre*ER *tdTom*fl/+ *Bcl9*fl/fl *Bcl9l*fl/fl (BCL9 BCL9l) mice injected with 0.15 mg tamoxifen and sampled 10 days later, blue = DAPI (nuclei) and red = tomato. Scale bar = 100 μm. **b** Average clone size from *Lgr5-EGFP-Cre*ER *tdTom*fl/+ (WT) and *Lgr5-EGFP-Cre*ER *tdTom*fl/+ *Bcl9*fl/fl *Bcl9l*fl/fl (BCL9 BCL9l) mice injected with 0.15 mg tamoxifen and sampled 4, 10 and 21 days post-induction. 200 clones per mouse were scored, $n = 3$ per group at each time point, one-way Mann–Whitney *U* test, *$P = 0.04$. **c** Heat map for % distribution of clone size from mice described in (**b**). Blue = low and red = high. Pie charts represent fixation of crypts—red = tomato labelled ISC, green = unlabelled ISC. **d** Serial sections of RFP and *Lgr5*-RNAscope staining on small intestines from *Lgr5-EGFP-Cre*ER *tdTom*fl/+ (WT) and *Lgr5-EGFP-Cre*ER *tdTom*fl/+ *Bcl9*fl/fl *Bcl9l*fl/fl (BCL9 BCL9l) mice sampled 21 days post-induction. Scale bar = 50 μm. Red arrows indicate RFP and *Lgr5* positive crypts, whilst blue arrows indicate RFP positive and *Lgr5* negative crypts. **e** Quantification of the number of tomato-positive clones/field of view from mice described in (**d**), sampled 21 days post-induction, $n = 3$ per group, one-way Mann–Whitney *U* test, $P = 0.04$. Data displayed as mean ±SEM

induce local recombination[40]. Mice were imaged via colonoscope at 2 and 4 weeks post-induction. Small tumours were visible in both groups 2 weeks post-induction; however after 4 weeks, tumours of *VillinCre*ER *Apc*fl/fl *Bcl9*fl/fl *Bcl9l*fl/fl mice were significantly smaller than those of *VillinCre*ER *Apc*fl/fl mice (Fig. 5g, h). This reduced rate of colonic tumour formation

translated into a survival benefit with *Bcl9/9l* deficiency (Supplementary Figure 6a). The colonic tumours that formed in *VillinCre*ER *Apc*fl/fl *Bcl9*fl/fl *Bcl9l*fl/fl mice retained expression of both *Bcl9* and *Bcl9l* (Supplementary Figure 6b), suggesting that BCL9/9l are required for colonic tumour growth following the loss of APC.

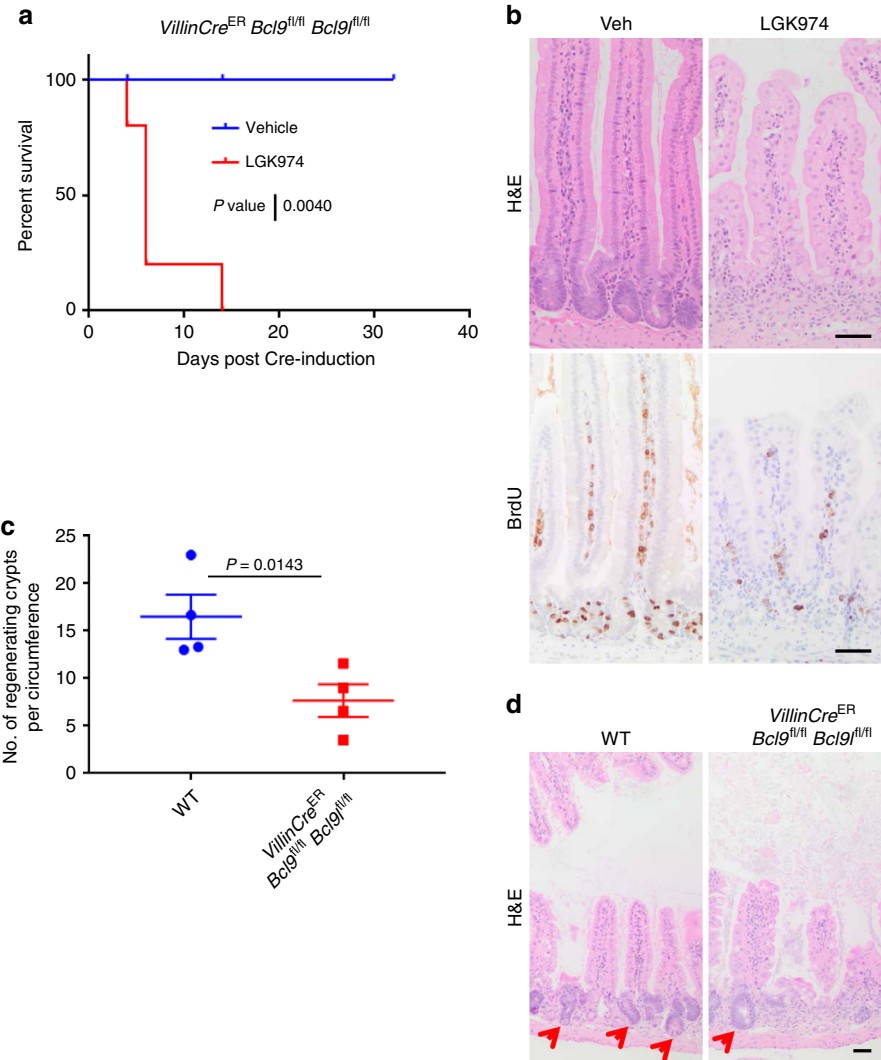

**Fig. 3** BCL9/9l are required for intestinal regeneration. **a** Survival plot for Cre-induced *VillinCre*[ER] *Bcl9*[fl/fl] *Bcl9l*[fl/fl] mice treated with 5 mg/kg LGK974 or vehicle, twice daily starting 24 h after tamoxifen injection, *n* = 5 per group. Log-rank test, *P*-value = 0.0040. **b** Representative H&E (upper panel) and BrdU (lower panel) staining of mice described in (**a**), sampled at end-point (LGK974) or at a time-point (vehicle). Scale bars = 50 μm. **c** Wildtype (WT) and *VillinCre*[ER] *Bcl9*[fl/fl] *Bcl9l*[fl/fl] mice were induced with tamoxifen and then 4 days post Cre-induction exposed to 10 Gy γ-irradiation and sampled 72 h later. The number of regenerating crypts per circumference in the small intestine was quantified and averaged per mouse, *n* = 4 per group, one-way Mann–Whitney *U* test, *P* = 0.0143. Data displayed as mean ±SEM. **d** Representative H&E stains of small intestines from mice described in (**c**). Red arrows indicate regenerating intestinal crypts. Scale bar = 50 μm

A possible explanation for the increased rate of tumour formation is that deletion of *Bcl9/9l* accelerated loss of the second copy of *Apc*. This could be achieved through DNA damage and loss of heterozygosity (LOH). We examined the abundance of γH2AX, a marker of DNA damage through IHC staining of intestinal tissue or tumours either proficient or deficient in BCL9/9l and found no difference (Supplementary Figure 7a). To examine LOH, we designed a high sensitivity in situ hybridisation probe (Basescope) to specifically detect exon 14 of *Apc*[17]. Given that this exon is specifically deleted in the *Apc*[fl] allele following Cre-mediated recombination, it follows that if the *Apc* locus undergoes LOH during tumour formation the probe will be undetected in resulting tumours. We observed that staining was negative in tumours, but positive in the adjacent normal epithelium, from both Cre-induced *VillinCre*[ER] *Apc*[fl/+] and *VillinCre*[ER] *Apc*[fl/+] *Bcl9*[fl/fl] *Bcl9l*[fl/fl] mice (Supplementary Figure 7b), suggesting that in both cases, the second copy of *Apc* is lost via LOH.

**BCL9/9l are required for mt*Ctnnb1* intestinal transformation.** The *Apc* allele used in the previous experiments is truncated at codon 580 and therefore lacks any β-catenin binding activity[41], whereas humans mutated APC frequently retains some binding. In the co-submitted article, Mieszczanek et al. recapitulate the increase in proximal intestinal tumour initiation following *Bcl9/9l* deletion in the SI of *Apc*[Min/+] mice—an allele which lacks β-catenin binding activity[42], while demonstrating that deletion almost completely blocks tumour formation in *Apc*[1322T/+] mice —an allele which retains some β-catenin binding activity. Hence, we hypothesised that the larger the APC protein and therefore more β-catenin binding activity that is retained, the more dependent cancer cells would become upon BCL9/9l for Wnt driven transformation.

To investigate this, we used a model that expresses a mutant form of β-catenin that cannot be degraded, resulting in intestinal transformation[15]. We hypothesised that in this instance, deletion of *Bcl9/9l* would slow transformation. Expression of a single copy

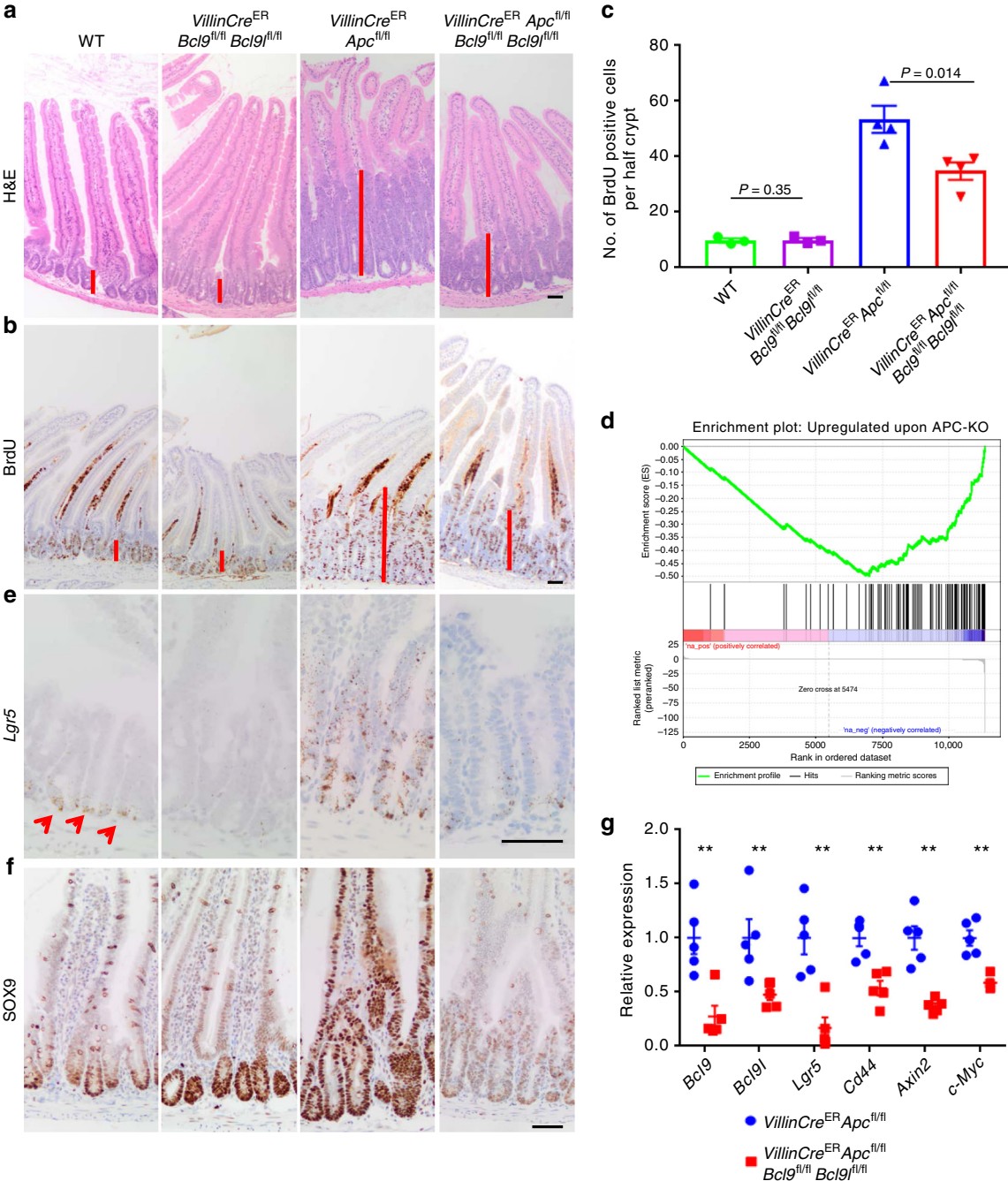

**Fig. 4** BCL9/9l are required for acute intestinal transformation. **a** Representative H&E staining of small intestines from WT, *VillinCre*^ER *Bcl9*^fl/fl *Bcl9l*^fl/fl, *VillinCre*^ER *Apc*^fl/fl and *VillinCre*^ER *Apc*^fl/fl *Bcl9*^fl/fl *Bcl9l*^fl/fl mice sampled 4 days post Cre-induction. Red bars indicate the size of the proliferative crypt. Scale bar = 50 μm. **b** Representative BrdU staining of small intestines from mice described in (**a**). Mice were injected intraperitoneally with BrdU 2 h prior to being culled. Red bars indicate the size of the proliferative crypt. Scale bar = 50 μm. **c** Quantification of proliferation (BrdU positive cells) in the small intestines of mice described in (**a**). The number of BrdU-positive cells per half crypt was quantified, 25 crypts per mouse scored, $n = 3$–4 for each group, one-way Mann–Whitney $U$ test, $P = 0.014$ for *VillinCre*^ER *Apc*^fl/fl vs *VillinCre*^ER *Apc*^fl/fl *Bcl9*^fl/fl *Bcl9l*^fl/fl and $P = 0.35$ for WT vs *VillinCre*^ER *Bcl9*^fl/fl *Bcl9l*^fl/fl. WT data from Supplementary Figure 2b. Data displayed as mean ±SEM. **d** Gene Set Enrichment Analysis of RNAseq data from small intestinal tissue from Cre-induced *VillinCre*^ER *Apc*^fl/fl and *VillinCre*^ER *Apc*^fl/fl *Bcl9*^fl/fl *Bcl9l*^fl/fl mice, $n = 3$ per group. **e** Representative *Lgr5*-RNAscope staining of small intestines from mice described in (**a**). Red arrows indicate *Lgr5*-staining at the base of WT crypts. Scale bar = 50 μm. **f** Representative SOX9 staining of small intestines from mice described in (**a**). Scale bar = 50 μm. **g** qPCR for Wnt target genes and intestinal stem cell markers from small intestinal tissue of Cre-induced *VillinCre*^ER *Apc*^fl/fl and *VillinCre*^ER *Apc*^fl/fl *Bcl9*^fl/fl *Bcl9l*^fl/fl mice, $n = 4$–5 per group, one-way Mann–Whitney $U$ test, **$P < 0.01$, $P = 0.00795$ (*Bcl9*), $P = 0.00395$ (*Bcl9l*), $P = 0.00395$ (*Lgr5*), $P = 0.00395$ (*Cd44*), $P = 0.00395$ (*Axin2*) and $P = 0.00795$ (*c-Myc*). Data displayed as relative to the mean of *VillinCre*^ER *Apc*^fl/fl mice. Data displayed as mean ±SEM

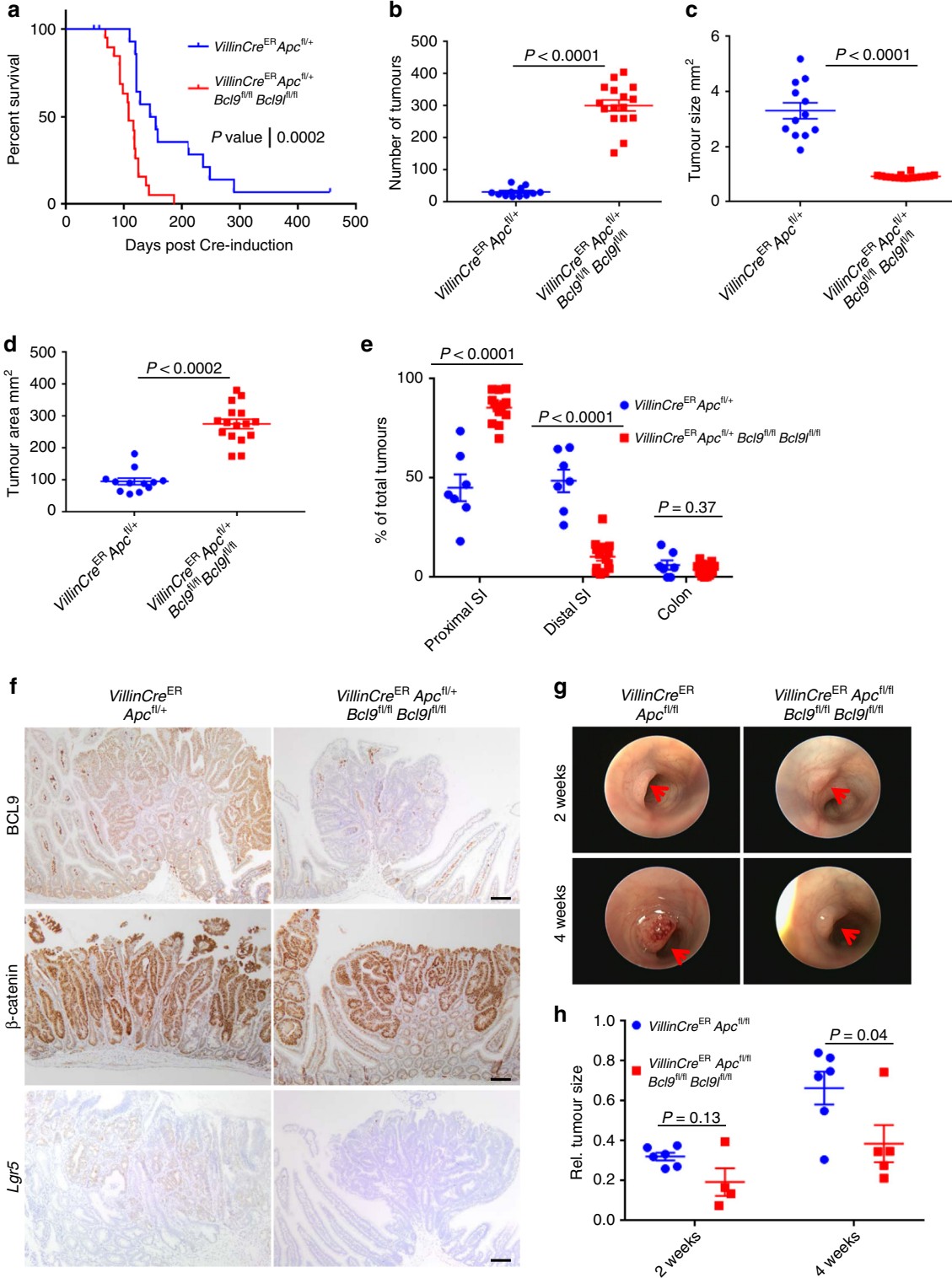

of mutant β-catenin ($Ctnnb1^{ex3/+}$) throughout the murine intestine and colon transformed the SI leading to a *crypt-progenitor phenotype* reminiscent of $Apc^{fl/fl}$ intestines, after approximately 21 days post Cre-induction. Deletion of *Bcl9/9l* significantly extended survival (Fig. 6a) and at endpoint, these mice had developed tumours as opposed to displaying the *crypt-progenitor phenotype* like the control cohort (Supplementary Figure 8a & b). Importantly, these tumours were escapers that retained expression of both *Bcl9* and *Bcl9l* (Supplementary Figure 8c & d). Sampling $VillinCre^{ER}$ $Ctnnb1^{ex3/+}$ $Bcl9^{fl/fl}$

$Bcl9l^{fl/fl}$ mice 21 days post Cre-induction confirmed that deletion of *Bcl9/9l* prevents the expansion of *Ctnnb1* mutant crypts. Intestinal crypts from Cre-induced $VillinCre^{ER}$ $Ctnnb1^{ex3/+}$ $Bcl9^{fl/fl}$ $Bcl9l^{fl/fl}$ mice are significantly smaller and exhibited a significant reduction in the number of BrdU positive cells when compared to $VillinCre^{ER}$ $Ctnnb1^{ex3/+}$ mice (Fig. 6b, c). There was a significant decrease in expression of a number of Wnt target genes including *Lgr5*, *Axin2*, SOX9 and CD44 in crypts from Cre-induced $VillinCre^{ER}$ $Ctnnb1^{ex3/+}$ $Bcl9^{fl/fl}$ $Bcl9l^{fl/fl}$ mice compared with Cre-induced $VillinCre^{ER}$ $Ctnnb1^{ex3/+}$ mice (Fig. 6c–e). This

**Fig. 5** Deletion of *Bcl9/9l* alters intestinal tumour distribution. **a** Survival curve for *VillinCre*[ER] *Apc*[fl/+] and *VillinCre*[ER] *Apc*[fl/+] *Bcl9*[fl/fl] *Bcl9l*[fl/fl] mice aged until clinical end-point, *n* = 16 for *VillinCre*[ER] *Apc*[fl/+] (3 censors—1 mouse had lymphoma, 1 with elongated teeth and another displayed rapid weight loss without significant tumour burden) and *n* = 19 for *VillinCre*[ER] *Apc*[fl/+] *Bcl9*[fl/fl] *Bcl9l*[fl/fl], Log-rank test, *P* = 0.0002. **b** Plot for total intestinal and colonic tumours from mice described in (**a**), *n* = 12 for *VillinCre*[ER] *Apc*[fl/+] and *n* = 16 for *VillinCre*[ER] *Apc*[fl/+] *Bcl9*[fl/fl] *Bcl9l*[fl/fl], one-way Mann–Whitney *U* test, *P* < 0.0001. Data displayed as mean ±SEM. **c** Plot for average tumour size from intestines and colons from mice described in (**a**), *n* = 12 for *VillinCre*[ER] *Apc*[fl/+] and *n* = 16 for *VillinCre*[ER] *Apc*[fl/+] *Bcl9*[fl/fl] *Bcl9l*[fl/fl], one-way Mann–Whitney *U* test, *P* < 0.0001. Data displayed as mean ±SEM. **d** Plot for total intestinal and colonic tumour burden from mice described in (**a**), *n* = 12 for *VillinCre*[ER] *Apc*[fl/+] and *n* = 16 for *VillinCre*[ER] *Apc*[fl/+] *Bcl9*[fl/fl] *Bcl9l*[fl/fl], one-way Mann–Whitney *U* test, *P* < 0.0002. Data displayed as mean ±SEM. **e** Plot of tumour distribution along the small intestine and colon of mice described in (**a**), *n* = 7 for *VillinCre*[ER] *Apc*[fl/+] and *n* = 14 for *VillinCre*[ER] *Apc*[fl/+] *Bcl9*[fl/fl] *Bcl9l*[fl/fl], one-way Mann–Whitney *U* test, *P* < 0.0001 (proximal SI and distal SI) and *P* = 0.37 for colon. Data displayed as % of total tumours/mouse. Data displayed as mean ±SEM. **f** Representative staining for BCL9 (top panel), β-catenin (middle panel) and *Lgr5*-RNAscope (bottom panel) of small intestinal tumours from mice described in (**a**). Scale bars = 100 μm. **g** Representative images from a colonoscopy of *VillinCre*[ER] *Apc*[fl/fl] and *VillinCre*[ER] *Apc*[fl/fl] *Bcl9*[fl/fl] *Bcl9l*[fl/fl] mice induced with a single injection of 4-hydroxytamoxifen into the colonic sub-mucosa. Red arrows indicate tumours. **h** Quantification of colonic tumour growth (normalised to the size of the lumen) of mice described in (**g**), *n* = 6 for *VillinCre*[ER] *Apc*[fl/fl] and *n* = 4–5 for *VillinCre*[ER] *Apc*[fl/fl] *Bcl9*[fl/fl] *Bcl9l*[fl/fl], one-way Mann–Whitney *U* test, *P* = 0.13 (2 weeks) and *P* = 0.04 (4 weeks). Data displayed as mean ±SEM

complete suppression of the mutant β-catenin-driven phenotype by deletion of *Bcl9/9l* was consistent with our proposed model in which oncogenic Wnt signalling above a threshold of Wnt activation is required for transformation.

To understand the contribution of BCL9/9l to the transformation of the SI driven by *Ctnnb1* mutations, we generated *VillinCre*[ER] *Ctnnb1*[ex3/ex3] *Bcl9*[fl/fl] *Bcl9l*[fl/fl] mice. These mice were induced and sampled 4 days post Cre-induction. *Bcl9/9l* deletion significantly reduced proliferation in both the SI and colon of Cre-induced *VillinCre*[ER] *Ctnnb1*[ex3/ex3] mice (Supplementary Figure S9a–d and Supplementary Figure 10a–c). Following transcriptional profiling of small intestinal tissue from Cre-induced *VillinCre*[ER] *Ctnnb1*[ex3/ex3] and *VillinCre*[ER] *Ctnnb1*[ex3/ex3] *Bcl9*[fl/fl] *Bcl9l*[fl/fl] mice, GSEA demonstrated that gene programmes upregulated following *Apc* deletion[39] or enriched in human CRC[43] are suppressed upon *Bcl9/9l* deletion (Supplementary Figure 9g and Supplementary Table 3). Moreover, *Bcl9/9l* deletion also significantly reduced the expression of a number of Wnt target genes, including *Lgr5*, *Axin2*, *Cd44* and *c-Myc* (Supplementary Figure 9e & f). The relative fold-change in expression of a subset of Wnt target genes including *Cd44* and *c-Myc* following *Bcl9/9l* deletion was more pronounced in the *VillinCre*[ER] *Ctnnb1*[ex3/ex3] setting (6- and 4.9-fold, respectively), compared to *VillinCre*[ER] *Apc*[fl/fl] (1.9- and 1.7-fold, respectively) (Supplementary Figure 11a & b). This supports the notion that cells which maintain a functional APC protein with significant β-catenin binding capacity have a higher dependency upon BCL9/9l for Wnt-driven transformation.

**BCL9/9l loss increases membrane β-catenin in mtCtnnb1 crypts** . To understand the relative dependencies of β-catenin mutant and APC-deficient tumours on BCL9/9l, we examined the intracellular distribution of β-catenin. Whilst in both cases we observed nuclear β-catenin staining (Figs. 5f and 6d) there was an increase in membranous β-catenin in crypts from *VillinCre*[ER] *Ctnnb1*[ex3/+] *Bcl9*[fl/fl] *Bcl9l*[fl/fl] mice sampled at 21 days post Cre-induction compared with Cre-induced *VillinCre*[ER] *Ctnnb1*[ex3/+] mice (Supplementary Figure 12a). From this we infer that the level of nuclear β-catenin is reduced following deletion of BCL9/9l. Interestingly, there was no difference in membrane-associated β-catenin in tumours from Cre-induced *VillinCre*[ER] *Apc*[fl/+] and *VillinCre*[ER] *Apc*[fl/+] *Bcl9*[fl/fl] *Bcl9l*[fl/fl] mice (Supplementary Figure 12b). This would argue that the larger the APC protein retained, the more dependent the cell is on BCL9/9l for the efficient shuttling of β-catenin from the cytoplasm and the membrane to the nucleus.

**BCL9/9l are required for mtCtnnb1 hepatocyte transformation**. Hepatocellular carcinoma (HCC) is characterised by activating mutations of β-catenin in approximately 30% of patients[44]. To investigate a role for BCL9/9l in liver transformation, we utilised an adeno-associated virus (AAV) system to express Cre recombinase specifically in hepatocytes to induce recombination of target genes[45]. We chose a titre of AAV8-TBG-Cre virus that gives near constitutive recombination of the adult liver. Acute deletion of *Bcl9/9l* did not perturb liver homeostasis or proliferation (Supplementary Figure 13a & b), though aged AAV8-TBG-Cre *Bcl9*[fl/fl] *Bcl9l*[fl/fl] mice did have a small yet significant reduction in liver-to-body weight ratio compared to controls (Supplementary Figure 13c). Moreover, we observed a reduced expression of the zonation marker and Wnt target gene Glutamine Synthetase after 140 days post-induction (Supplementary Figure 13a). We investigated the role for BCL9/9l in β-catenin-driven hepatocyte transformation through generation of *Ctnnb1*[ex3/+] and *Ctnnb1*[ex3/+] *Bcl9*[fl/fl] *Bcl9l*[fl/fl] mice and treated with AAV8-TBG-Cre virus. Acute hepatic expression of a single copy of mutant β-catenin leads to an increase in the expression of a number of Wnt target genes, including *Lgr5*, *Axin2*, *Bcl9* and *Bcl9l* 4 days post-induction, in a BCL9/9l dependent manner (Supplementary Figure 14a–d). AAV8-TBG-Cre *Ctnnb1*[ex3/+] mice develop hepatomegaly within 2 weeks post-induction, and develop a liver failure phenotype which was completely suppressed through deletion of *Bcl9/9l*, where mice were aged up to 140 days before being euthanised (Fig. 7a and Supplementary Figure 14e). Interestingly after 140 days these mice did exhibit small liver lesions, deficient for *Bcl9/9l* expression (Supplementary Figure 14f).

Given the survival extension provided by deletion of *Bcl9/9l*, we chose to compare livers from AAV8-TBG-Cre *Ctnnb1*[ex3/+] and AAV8-TBG-Cre *Ctnnb1*[ex3/+] *Bcl9*[fl/fl] *Bcl9l*[fl/fl] mice sampled at day 14. Here, there was a significant reduction in liver-to-body weight ratio of AAV8-TBG-Cre *Ctnnb1*[ex3/+] *Bcl9*[fl/fl] *Bcl9l*[fl/fl] mice compared with AAV8-TBG-Cre *Ctnnb1*[ex3/+] mice (Fig. 7b). The reduced liver-to-body weight ratio was also accompanied by a significant reduction in the number of BrdU positive hepatocytes (Fig. 7c). We also observed downregulation of a number of Wnt targets, including SOX9, *Lgr5* and *Axin2* following deletion of *Bcl9/9l* (Fig. 7d). Additionally, mutant β-catenin drove the expansion of the Glutamine Synthetase positive zone around the central vein; which was suppressed by deletion of *Bcl9/9l* (Fig. 7d).

## Discussion

Since the finding that restoration of APC expression can cause regression of aggressive CRC, there has been renewed interest in

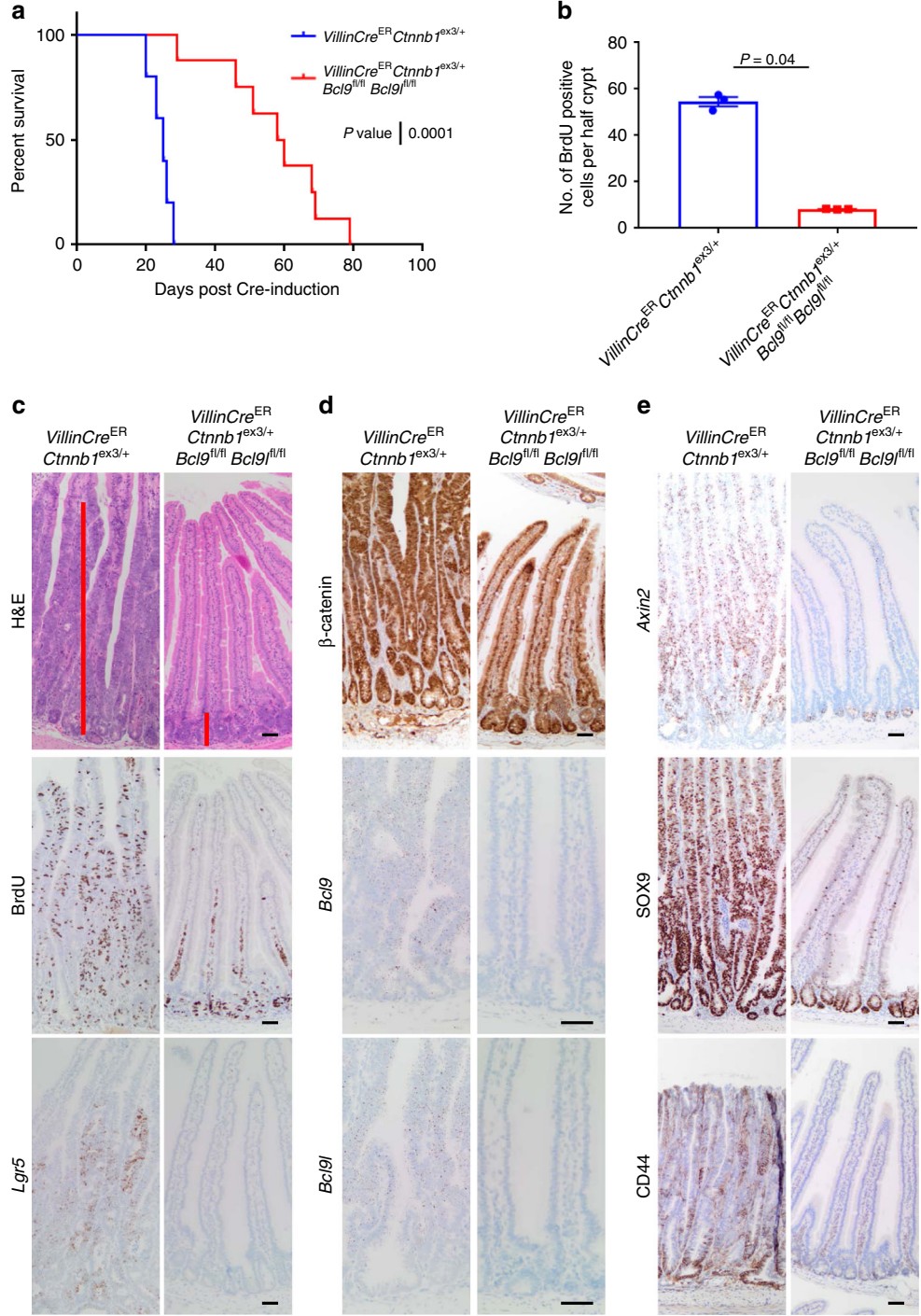

**Fig. 6** BCL9/9l are required for mt*Ctnnb1* intestinal transformation. **a** Survival curve for Cre-induced *VillinCre*ER *Ctnnb1*ex3/+ and *VillinCre*ER *Ctnnb1*ex3/+ *Bcl9*fl/fl *Bcl9l*fl/fl mice aged until clinical endpoint, n = 5 for *VillinCre*ER *Ctnnb1*ex3/+ and n = 8 for *VillinCre*ER *Ctnnb1*ex3/+ *Bcl9*fl/fl *Bcl9l*fl/fl, log-rank test, P = 0.0001. **b** Quantification of proliferation (BrdU positive cells) in the small intestines of Cre-induced *VillinCre*ER *Ctnnb1*ex3/+ sampled at end-point and *VillinCre*ER *Ctnnb1*ex3/+ *Bcl9*fl/fl *Bcl9l*fl/fl mice sampled 21 days post Cre-induction. The number of BrdU-positive cells per half crypt was quantified, 25 crypts per mouse scored, n = 3 for each group, one-way Mann–Whitney U test, P = 0.04. Data displayed as mean ±SEM. **c** Representative H&E (upper panels— red bar indicates the size of the proliferative zone), BrdU (middle panels) and *Lgr5*-RNAScope (lower panels) staining of small intestinal sections from Cre-induced *VillinCre*ER *Ctnnb1*ex3/+ (at end-point) and *VillinCre*ER *Ctnnb1*ex3/+ *Bcl9*fl/fl *Bcl9l*fl/fl (sampled at day 21) mice. Mice were injected with BrdU intraperitoneally 2 h prior to being culled. Scale bars = 50 µm. **d** Representative β-catenin (upper panel), *Bcl9*-RNAScope (middle panel) and *Bcl9l*-RNAScope (lower panel) staining of small intestinal sections from mice described in (**b**). Scale bars = 50 µm. **e** Representative *Axin2*-RNAScope (upper panel), SOX9 (middle panel) and CD44 (lower panel) staining of small intestinal sections from mice described in (**b**). Scale bars = 50 µm

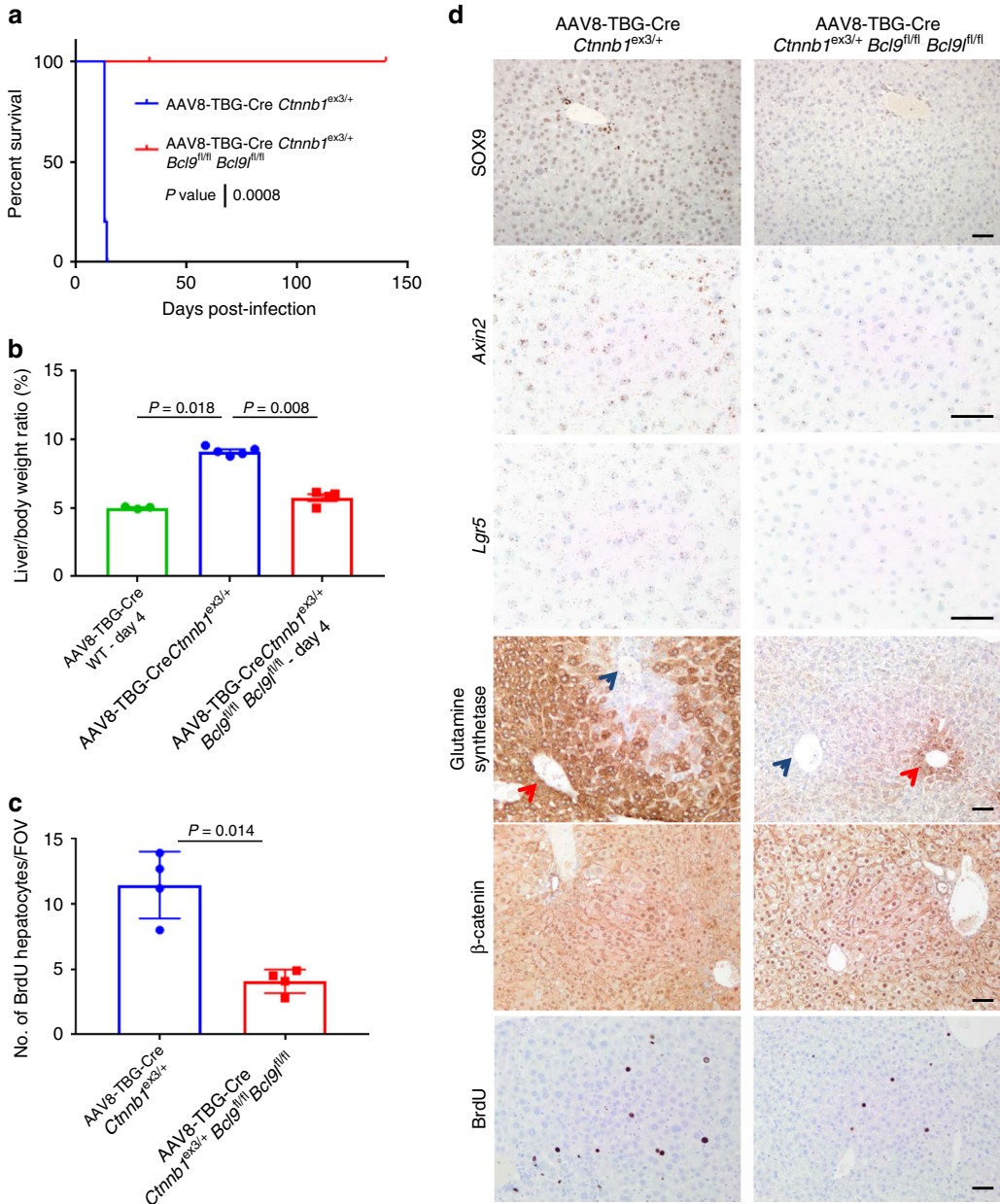

**Fig. 7** BCL9/9l are required for mtCtnnb1 hepatocyte transformation. **a** Survival curve for AAV8-TBG-Cre induced *Ctnnb1*ex3/+ and *Ctnnb1*ex3/+ *Bcl9*fl/fl *Bcl9l*fl/fl mice aged until clinical end-point or time-point, n = 5 per group, Log-rank test, P = 0.0008. **b** Liver-to-body weight ratio (%) of AAV8-TBG-Cre induced WT (sampled at day 4), *Ctnnb1*ex3/+ (sampled at end-point) and *Ctnnb1*ex3/+ *Bcl9*fl/fl *Bcl9l*fl/fl (sampled at day 14), n = 3–5 per group, one way Mann–Whitney U test, P = 0.018 AAV8-TBG-Cre WT vs AAV8-TBG-Cre *Ctnnb1*ex3/+ and P = 0.008 for AAV8-TBG-Cre *Ctnnb1*ex3/+ vs AAV8-TBG-Cre *Ctnnb1*ex3/+ *Bcl9*fl/fl *Bcl9l*fl/fl. Data displayed as mean ±SEM. **c** Quantification of BrdU positive hepatocytes from AAV8-TBG-Cre *Ctnnb1*ex3/+ (sampled at end-point) and *Ctnnb1*ex3/+ *Bcl9*fl/fl *Bcl9l*fl/fl (sampled at day 14). Number of BrdU positive hepatocytes scored per ×20 objective field of view (FOV), 10 FOVs scored per mouse, n = 4 per group. One-way Mann–Whitney U test, P = 0.014. Data displayed as mean ±SEM. **d** Representative staining for SOX9, *Axin2*-RNAscope, *Lgr5*-RNAscope, Glutamine Synthetase (red arrows indicate the central vein and blue arrows indicate the portal tract areas, respectively), β-catenin and BrdU in liver sections from AAV8-TBG-Cre induced *Ctnnb1*ex3/+ and *Ctnnb1*ex3/+ *Bcl9*fl/fl *Bcl9l*fl/fl mice aged until clinical end-point or time-point. Scale bar = 50 μm

targeting Wnt signalling in APC deficient CRC[24]. Although an excellent proof of concept, there is a paucity of strategies to target the Wnt pathway in ligand independent cancers that have mutations in either *APC* or *CTNNB1*. Our work highlights the possibility that inhibiting BCL9/9l could provide an excellent strategy. Moreover, we elucidate key threshold levels of Wnt signalling which differentiate normal homeostasis from transformation. Most importantly, our study and the co-submitted study by Mieszczanek et al. highlight that the *APC* mutations

associated with human CRC that retain β-catenin binding will be most sensitive to BCL9/9l inhibition.

Key among our conclusions is the elucidation of two different transcriptional programmes driven by BCL9/9l during homeostasis and transformation. We find that BCL9/9l are dispensable for intestinal homeostasis, although required for the expression of the ISC marker Lgr5 as previously suggested[29]. Importantly not only did acute deletion of *Bcl9/9l* lead to the loss of the Lgr5+ ISC gene signature, it functionally reduced ISC fitness compared to

WT ISCs. This perturbation in the Lgr5+ ISC pool following *Bcl9/9l* deletion may explain why BCL9/9l-null intestinal crypts have reduced capacity to regenerate following irradiation since Lgr5+ ISCs are required for intestinal regeneration[46]. We observed that there was a significant increase in the E-cadherin bound β-catenin following deletion of *Bcl9/9l*, agreeing with previous studies highlighting a role for these two proteins in the nuclear shuttling of β-catenin.

Despite hyperactive Wnt signalling being a hallmark of CRC, efficacious therapies against the pathway are limited. This is due to the lack of agents that act downstream of the destruction complex. As BCL9/9l are required for a subset of β-catenin-mediated transcriptional targets in the normal intestine and are dispensable for intestinal homeostasis we investigated whether they play a role in intestinal epithelial transformation. Acute deletion of both copies of *Apc* rapidly transforms the murine intestine within 4 days[39]. This hyperproliferative phenotype along with the expression of a large number of Wnt target genes was suppressed following deletion of *Bcl9/9l*. GSEA revealed that BCL9/9l are required for the expression of a transcriptional programme that is induced following APC loss in the mouse intestine[39] and Wnt target genes that are upregulated in human CRC[43]. This suggests that BCL9/9l are required for an oncogenic β-catenin-mediated transcriptional programme which permits the acute transformation of the murine intestine following APC loss. It is important to note that with the exception of *Lgr5*, many Wnt target genes that are upregulated following APC loss, including *Axin2*, *Cd44* and *c-Myc* were specifically reduced when *Bcl9/9l* were deleted concurrently with *Apc*, as opposed to in the WT intestinal epithelium where these genes were unaffected.

A key observation from the tumour models driven by APC loss was that deletion of *Bcl9/9l* favoured adenoma formation within the proximal SI, whilst colonic tumour growth was suppressed. It has previously been proposed that human CRC tumours have a 'just-right' level of Wnt signalling, which may in fact be sub-maximal due to truncated APC proteins that retain β-catenin binding sites[13,14]. Furthermore, a decreasing Wnt gradient has been described from the proximal SI to the distal colon[15]. Together, the basal level of Wnt signalling within the intestinal epithelium along with intra-tumoural Wnt signalling has been proposed to influence the regional distribution of tumours in both humans and mice. For instance, $Apc^{Min/+}$ and $Apc^{1322T/+}$ mice, which harbour different truncating *Apc* mutations, have distinct tumour distributions along their SI due to the different levels of Wnt signalling within their tumours[15,42]. Tumours from $Apc^{Min/+}$ mice have high levels of Wnt signalling which is not permissive for proximal SI tumour formation; hence tumours form in the distal SI. Conversely, $Apc^{1322T/+}$ tumours retain two β-catenin binding domains and consequently have sub-maximal Wnt signalling, permitting tumour formation in the proximal intestine[15,42]. This in turn may explain the tumour distribution we observe in $VillinCre^{ER}$ $Apc^{fl/+}$ $Bcl9^{fl/fl}$ $Bcl9l^{fl/fl}$ mice, since our $Apc^{580S}$ allele lacks any β-catenin binding sites[41]. Here deletion of *Bcl9/9l* reduces intratumoural Wnt signalling, favouring tumour formation in the proximal SI, but is not permissive for colonic tumour formation due to the relatively low underlying basal Wnt signalling; observations which support the 'just-right' Wnt signalling hypothesis. Here our data is very consistent with the submission of Mieszczanek et al. They show that the $Apc^{Min/+}$ mouse also develops a large number of small proximal intestinal tumours when *Bcl9/9l* is deleted, whilst $Apc^{1322T/+}$ mice are resistant to tumorigenesis. Therefore loss of BCL9/9l now reduces Wnt signalling activation to a level that can no longer transform the intestine. Our parallel studies using an activated mutant β-catenin allele supports this hypothesis. Here there is almost a complete abrogation of both intestinal and liver transformation

driven by mutant β-catenin when BCL9/9l are lost. Interestingly, BCL9 expression has been shown to increase as HCC progresses and that those patients with high BCL9 expression have a worse prognosis[47,48], highlighting the human relevance of our findings.

Whilst the data presented here are consistent with the 'just-right' hypothesis of Wnt signalling, it is important to note it remains correlative. Other cell-extrinsic factors such as niche factors, the microbiome or nutrient availability vary greatly throughout the length of the intestine and these in turn may impact the regional distribution of tumours within $VillinCre^{ER}$ $Apc^{fl/+}$ $Bcl9^{fl/fl}$ $Bcl9l^{fl/fl}$ mice. Therefore, further studies are required to confirm the 'just-right' hypothesis. It is notable that upon comparison of gene expression profiles following *Apc* deletion to those following expression of two copies of mutant β-catenin concomitant with loss of BCL9/9l, we find that while some canonical Wnt targets such as *Axin2* and *Lgr5* are equally downregulated, others, including *c-Myc* and *Cd44* are more substantially reduced following β-catenin mutation. This demonstrates that in the context of specific Wnt activating mutations, BCL9/9l loss preferentially affects specific Wnt target genes, such as *c-Myc*, which in turn may be crucial for transformation.

Mechanistically our data supports a model that in normal cells, BCL9/9l is redundant for most Wnt target genes and is only required for genes dependent upon the highest level of Wnt signalling, such as Lgr5+ ISC genes. This is consistent with work using Wnt inhibitors in vivo, where the ISC genes are the most sensitive to Wnt inhibition[35]. Following the loss of BCL9/9l there is more β-catenin at cell junctions and less β-catenin available for optimal target gene expression of ISC genes. Following APC loss there is a global role for BCL9/9l in the Wnt enhanceosome to allow optimal expression of Wnt target genes. In the colon, reducing expression of these target genes is sufficient to suppress tumorigenesis. In the Wnt high proximal SI this level of Wnt is now very efficient for tumour initiation. However, if the truncated APC protein can still bind β-catenin, this leads to a further reduction of Wnt signalling with more β-catenin at cell junctions and a concomitant reduction of Wnt target gene expression. This leads to inefficient tumorigenesis in either $Apc^{1322T/+}$ or $Ctnnb1^{ex3/+}$ mice (Fig. 8). These observations raise the possibility that reduction rather than ablation of Wnt signalling can be efficacious in human tumours carrying these mutations.

In summary, we have uncovered a role for BCL9/9l downstream of the β-catenin destruction complex, mediating a β-catenin-driven oncogenic transcriptional programme required for Wnt-mediated intestinal and hepatocyte transformation. Continued development of BCL9/9l inhibitors may yield a therapeutic window for CRC and β-catenin-driven HCC.

## Methods

**Mouse experiments**. *Mouse colonies*: All experiments were performed according to UK Home Office regulations (licence 70/8646), and reviewed by local ethical review committee at the University of Glasgow. Male and female C57BL/6J >20 g mice were induced with tamoxifen from 6 to 12 weeks of age. The alleles used were as follows: $VillinCre^{ER}$ [49], $Apc^{580S}$ [41], $Bcl9^{fl}$, $Bcl9l^{fl}$ [28], $Lgr5Cre^{ER}$ [50], $R26R$-LSL-tdTomato ($tdTom^{fl}$)[51] and $Ctnnb1^{ex3}$ [52]. Recombination in the acute models was induced using a single intraperitoneal injection of 80 mg/kg tamoxifen for 2 consecutive days and mice sacrificed 4 days post-induction. $VillinCre^{ER}$ $Apc^{fl/+}$ mice were aged until they showed clinical signs (anaemia, hunching and/or weight loss).

Intracolonic Cre inductions were administered under general anaesthesia—a single 70 μl 100 nM dose of 4-hydroxy tamoxifen (Merck Millipore, Cat# 579002-5MG) was injected into the colonic sub-mucosa via a colonoscope. Colonic tumour growth was then monitored via a colonoscope. Tumour volume was measured relative to lumen size using ImageJ.

For regeneration experiments, mice were exposed to γ-irradiation from a caesium-137 source. This delivered γ-irradiation at 0.423 Gy min$^{-1}$.

The Porcupine inhibitor LGK974 was administered in a concentration of 5 mg/kg BID (oral gavage) in a vehicle of 0.5% Tween-80/0.5% methylcellulose.

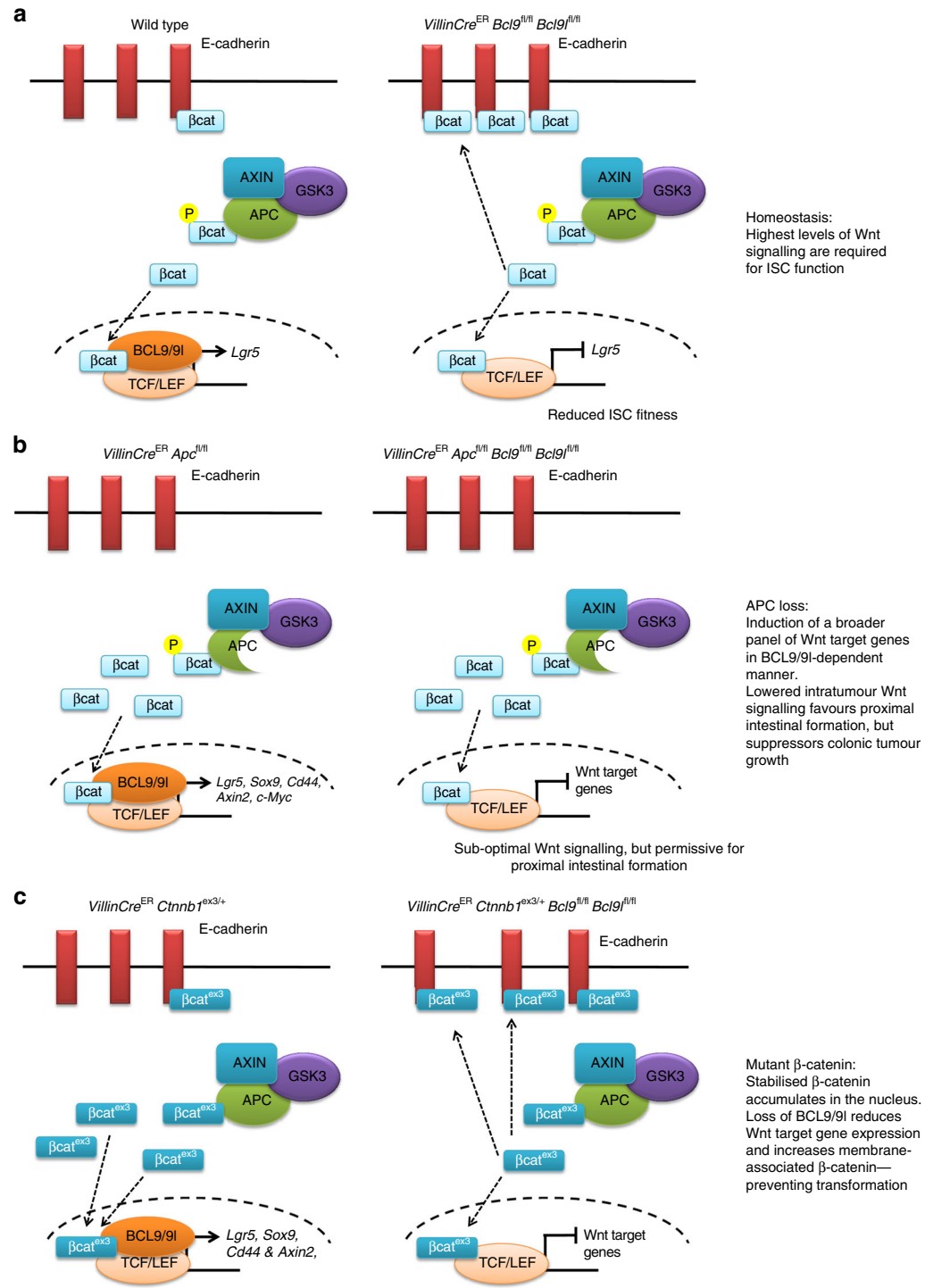

**Fig. 8** BCL9/9l play a context-dependent role in β-catenin mediated transcription. **a** In the WT setting, BCL9/9l are required for *Lgr5* transcription and expression of the *Lgr5*-ISC signature. Upon deletion of *Bcl9/9l*, there is increased E-cadherin bound β-catenin and a reduction in ISC fitness. **b** Following the loss of APC, there is an increase in nuclear β-catenin and concomitant increase in the expression of a number of Wnt target genes in a BCL9/9l-dependent manner, such as *Lgr5*, *Axin2*, *Sox9*, *Cd44* and *c-Myc*. Deletion of *Bcl9/9l* significantly reduces the expression of many of these Wnt target genes, however this global reduction in Wnt signalling is permissive for tumour formation in the proximal small intestine, but not in the colon. **c** Expression of a mutant copy of β-catenin that can no longer be phosphorylated induces a Wnt transcriptional programme even in the presence of an intact destruction complex. Upon deletion of *Bcl9/9l*, there is increased membrane-associated β-catenin and a subsequent failing to induce a transcriptional programme that is sufficient mutant for β-catenin-driven intestinal and hepatocyte transformation

AAV mediated recombination was performed as previously described[53]. Briefly, viral particles ($2 \times 10^{11}$ genetic copies/mouse) of AAV8.TBG.PI.Cre.rBG (UPenn Vector Core, Catalogue number: AV-8-PV1091) were injected via tail vein in 100 µl PBS. Mice were sacrificed and analysed at the indicated timepoints or aged until clinical endpoint—weight loss, hunching and a swollen abdomen.

In accordance with the 3Rs, the smallest sample size was chosen that could give a significant difference. Given the robust phenotypes of the $Apc^{fl/fl}$ model, and our prediction that BCL9 and BCL9l were essential, the minimum sample size assuming no overlap in control vs. experimental is three animals. No randomisation was used and the experimenter was blinded to genotypes.

**Immunohistochemistry**. IHC was performed on formalin-fixed intestinal sections. Standard IHC techniques were used throughout this study. Primary antibodies used for immunohistochemistry were as follows: BrdU (1:200, BD Biosciences #347580), SOX9 (1:500, Chemicon #AB5535), β-catenin (1:50, BD Biosciences #610154), BCL9 (1:500, Abnova #H00000607-MO1), Glutamine Synthetase (1:200 BD Biosciences #610518), γ-H2AX (1:50, Cell Signalling Technologies #9718) and CD44 (1:50 BD Biosciences #550538). For each antibody, staining was performed on at least three mice of each genotype, representative images are shown for each staining. For nuclear β-catenin staining Tris-EDTA based antigen retrieval was used.

**Immunofluorescence**. IF was performed on formalin-fixed intestinal sections. Citrate buffer antigen retrieval was used on all sections. Primary antibodies used for IF were as follows: β-catenin (1:200, BD Biosciences #610154) and E-cadherin (1:200, Cell Signalling technologies #3195). Sections were stained with DAPI before mounting. Sections were imaged on Zeiss LSM confocal microscope with a ×40 objective.

**RNAscope**. In situ hybridisation detection for $Lgr5$ (312178), $Olfm4$ (311838), $Axin2$ (400338), $Bcl9$ (529268) and $Bcl9l$ (466698) mRNA (All Advanced Cell Diagnostics) was performed using RNAscope 2.5 LS (Brown) Detection Kit (Advanced Cell Diagnostics) on a Bond Rx autostainer (Leica) strictly according to the manufacturer's instructions. Basescope (Advanced Cell Diagnostics) ApcEx14 #701641 (detects wild-type $Apc$ exon 14) was used according to the manufacturer's instructions.

**Proliferation and regeneration**. Proliferation levels were assessed by measuring BrdU incorporation. Mice were injected with 250 µl of BrdU (Amersham Biosciences) 2 hours before being sacrificed. Immunohistochemical staining for BrdU was then performed using an anti-BrdU antibody. For each analysis, 25 half crypts were scored per mouse from at least three mice of each genotype. Regenerating crypts were scored from H&E-stained sections as previously described[54]. The number of regenerating crypts per circumference of a small intestinal section was scored from at least 5 different sections per mouse. A minimum of three mice per genotype was scored. For liver sections, the number of BrdU positive fields of view were quantified, at least 10 views were scored per mouse from at least three mice of each genotype.

**Quantitative PCR (qRT-PCR)**. Whole pieces of small intestinal tissue or liver were used for RNA purification using RNeasy Mini Kit (QIAGEN, #74104) according to the manufacturer's instructions. 1 µg of RNA was reverse transcribed using DyNAmo cDNA Synthesis Kit (Thermo Scientific, #F-470L) according to the manufacturer's instructions, cDNA was diluted 1:10 in RNase-free water. qPCR was performed on each sample in technical duplicate, and with at least three biological replicates per genotype, in a 20 µl reaction mixture containing 10 µl of 2XDyNAmo HS master mix (Thermo Scientific), 0.5 µM of each of the primers (detailed later) and 3 µl cDNA generated previously. The reaction mixture without a template was run in duplicate as a control. The reaction conditions were as follows: 95 °C for 15 min, followed by 40 cycles of three steps consisting of denaturation at 95 °C for 15 s, primer annealing at 60 °C for 30 s, and primer extension at 72 °C for 30 s. A melting curve analysis was performed from 65 to 95 °C in 0.5 °C intervals. $Gapdh$ was used to normalise for differences in RNA input. Primer sequences are described in Supplementary Table 4.

**Crypt culture**. Mouse small intestines were isolated from wildtype and Cre-induced $VillinCre^{ER}$ $Bcl9^{fl/fl}$ $Bcl9l^{fl/fl}$ mice sacrificed 4 days post tamoxifen injection, and opened longitudinally and washed with PBS. Crypts were isolated as previously described[55]. Isolated crypts were mixed with 20 µl of Matrigel (BD Bioscience), plated in 24-well plates in Advanced DMEM/F12 supplemented with penicillin–streptomycin, 10 mM HEPES, 2 mM glutamine, N2, B27 (all from Gibco, Life Technologies), 100 ng ml$^{-1}$ Noggin and 50 ng ml$^{-1}$ EGF (both Peprotech). Wild-type crypts were also supplemented with R-spondin conditioned medium. Growth factors were added every 2 days.

**RNAseq**. Whole tissue from the small intestine was used for RNA purification. RNA integrity was analysed with a NanoChip (Agilent RNA 6000 Nanokit #5067-1511). A total of 2 µg of RNA was purified via Poly-A selection. The libraries were run on the Illumina Next Seq 500 using the High Output 75 cycles kit ($2 \times 36$ cycles, paired-end reads, single index). Analysis of the RNAseq data was carried out as previously described in ref. [56].

**Gene set enrichment analysis (GSEA)**. GSEA analysis was performed using the GSEA v2.0 software (Broad Institute). The comparison gene sets were obtained from published sources; $Lgr5$ high vs low study A, $Lgr5$ cross-platform[57], progenitor cluster[58], genes upregulated following APC-KO[39], Wnt target genes that are increased in human CRC[43], direct and functional β-catenin targets in SW480 cells[59].

**Proximity ligation assay (PLA)**. PLA was performed on tissue samples fixed at 4 °C for <24 h in 10% formalin prior to processing using the Duolink Detection kit (Sigma) according to the manufacturer's instructions. Briefly, after citrate buffer-mediated antigen retrieval, the slides were incubated with goat E-cadherin (1:200, R&D Systems AF748) and mouse β-catenin (1:2000, #610154, BD Biosciences) overnight. Detection was performed with PLA probes (anti-goat and anti-mouse) conjugated to oligonucleotides. After ligation, amplification detection with a fluorescent probe, slides were imaged on a Zeiss LSM confocal microscope. Z-stacks with ×40 objectives were taken. PLA dots in crypts were analysed with ImageJ calculated as area fraction.

**Clonal counting**. $Lgr5$-EGFP-Cre$^{ER}$ $tdTom^{fl/+}$ and $Lgr5$-EGFP-Cre$^{ER}$ $Bcl9^{fl/fl}$ $Bcl9l^{fl/fl}$ mice were induced with 0.15 mg tamoxifen and then aged for 4, 10 and 21 days. The proximal small intestine was isolated flushed and opened longitudinally and then fixed in 4% paraformaldehyde at room temperature for 4 hours. Tissue was subsequently stored in PBS at 4 °C ahead of processing. Tissue was incubated with DAPI overnight and imaged lumen side down using a Zeiss LSM confocal microscope on a ×10 objective. On average 20 images were acquired from each mouse and the proportion of Tomato positive crypts was determined and a minimum of 200 crypts scored per mouse.

**Liver enzyme detection**. Plasma was separated from blood obtained via cardiac puncture after collection into heparin. Liver biochemistry was performed using the Siemens Dimension Expand clinical chemistry system and compatible kits (Siemens Diagnostics, USA).

**Statistical analysis**. Statistical analysis was performed with GraphPad Prism V6 Software (La Jolla, CA, USA) using one-tailed Mann–Whitney tests or otherwise stated. For individual value plots, data displayed as mean ±standard error of the mean (SEM).

**Reporting summary**. Further information on experimental design is available in the Nature Research Reporting Summary linked to this article.

## Data availability
The authors declare that all relevant data supporting the findings of this study are available within the article and its Supplementary Information files. RNAseq data that support the findings of this study have been deposited in the ArrayExpress database under accession number E-MTAB-7546. Additional information can be obtained from the corresponding author (O.J.S.).

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

## Acknowledgements

O.J.S. is supported by Cancer Research UK grants (C596/A17196, A12481 and A21139) and an ERC starting grant (311301). D.M.G. is supported by an ERC starting grant (311301). T.G.B. and M.M. are supported by a Wellcome Trust grant (WT107492Z). R.J. is supported by a European Research Council MSCA fellowship (659666). M.C.H. is supported by a MRC doctoral training grant (MR/J50032x/1). J.D.L. is supported by a MRC Clinical Research Training Fellowship (MR/N021800/1). The authors thank CRUK Beatson Institute's transgenic team and Biological Service Unit for all help with mouse derivation and husbandry. Thanks also to Margaret O'Prey and David Strachan for help with confocal imaging.

## Author contributions

O.J.S and D.M.G. designed the project. D.M.G., R.A.R. and M.M. performed breeding and phenotypic analysis of mice. A.H. and W.C. performed RNAseq analysis. M.C.H. performed Gene Set Enrichment Analysis. R.J., J.D.L., M.M., D.J.H., C.N., A.D.C. and T.G.B. provided advice and material. D.M.G. and O.J.S. wrote and edited the manuscript.

**Additional information**

**Competing interests:** The authors declare no competing interests.

