## [Peer Review File · Nature Communications]

Reviewers' comments:

Reviewer #1 (Remarks to the Author):

In the study "Loss of BCL9/9I strongly suppresses Wnt driven tumourigenesis and functionally supports the "just right" Wnt signalling hypothesis" by Prof Sansom and colleagues, the role of BCL9/9L in intestinal homeostasis and regeneration, and tumor formation in the intestine and liver is investigated. It is unequivocally demonstrated that BCL9/9L is required for maintenance of Wnt signaling levels required to sustain Lgr5+ ISCs and drive intestinal regeneration. Furthermore, interesting differences are observed regarding the effect of Bcl9/9I deletion on tumor formation following Apc-loss in the various segments of the bowel. These observations are intriguing but too much emphasis is put on the relationship with Wnt activity levels and various Apc mutations, and just-right signaling. To conclude, the effect of BCL9/9I loss is investigated in beta-catenin driven tumorigenesis in the liver. Altogether the work provides an important advance in the field and is well written albeit with some minor textual inaccuracies. I support publication if some aspects of the work have been adapted.

Concerns:

1. The results of the proximity ligation assay Beta-cat E-cadherin I did not find particularly convincing. Figure 1B should be improved as I do not see a difference when the quantification indicates it should be there.
2. The clonal analysis presented in Fig 2A-C is an interesting highlight of the work and indicates an important impact on stem cell population dynamics of Bcl9/9I. This analysis would be strengthened in case the overlap in allele floxing of tdtom, Bcl9 and Bcl9L was directly demonstrated. Furthermore, is there a reduced number of clones detected in Bcl9/9I --/-- condition?
3. The finding that Bcl9/9I loss increases adenoma formation in the proximal small intestine and reduces it in the distal small intestine is very interesting. Similarly, the accompanying manuscript discusses an noteworthy observation in which different Apc truncating mutations present with different phenotypes in presence or absence of Bcl9. Yet the interpretation of the authors that this is related to 'just right' signaling and basal levels of Wnt activity at the various compartments of the intestine, and different effects of specific mutations, is simply not indisputably proven. There is lots of hand waving in the text to make this point, but evidently the proper (admittedly very complex) experiments to address this hypothesis have not been performed. The current argument hinges on circumstantial evidence and comparison of multiple model systems, and various segments of the intestine that differ in many more aspects than only basal Wnt levels (or specific Apc truncating mutation). For example it might be related to different dependencies of niche factors, different apoptosis thresholds, etc. I propose that the authors clearly indicate that the current interpretation is as speculative as any other possible explanation at this stage and adapt the text, title and abstract to reflect this.
4. I wonder if with the basescope assay for exon 14 of Apc LOH can be proven. Why can't other loss events be more common?
5. Few grammatical errors and typos are present in the text, e.g.:
line 79: 'The beta-catenin-TCF interface is a large and dynamic with...'
line 105: ' BCL/9I'
etc.

Reviewer #2 (Remarks to the Author):

In this comprehensive in vivo study, Gay et al examined the function of Bcl9 and Bcl9l, key components Wnt enhanceosome, in intestinal stem cell fitness, intestinal regeneration, and tumorigenesis. Using conditional Bcl9/9l KO mice, authors showed that BCL9/9l are required for the fitness of intestinal stem cells, for intestinal regeneration, but not for normal tissue homeostasis of the intestine. Loss of Bcl9/9l suppressed APC loss-induced crypt progenitor phenotype with strong reduction of beta-catenin target gene expression. Authors further showed that loss of Bcl9/9l strongly suppressed mutant beta-catenin-driven intestinal transformation and hepatocyte transformation. Authors also made an interesting observation that Bcl9/9l knockout promoted proximal tumor initiation, which functionally supports the just right hypothesis. Targeting beta-catenin signaling in cancer is complicated by the function of beta-catenin in normal tissue homeostasis, and it is not clear how to generate a therapeutic window. Authors have generated a compelling set of data suggesting that Bcl9/9l are required for high beta-catenin signaling involved in Lgr5+ ISC function and cell transformation, but not for low beta-catenin signaling involved in normal intestinal tissue homeostasis. The study provides strong genetic validation of targeting Bcl9/9l in cancers with APC or beta-catenin mutation. This is especially interesting considering recent findings that Lgr5+ cells might function as tumor stem cells in intestinal tumors. Although the impact of Bcl9 inhibition on other tox tissues such as the bone is still not clear and the study has not touched the function of Bcl9 in tumor maintenance, this work represents a significant step forward and is of importance for the Wnt field. Experiments are beautifully done and data are convincing. It certainly merits publication in Nature Communication.

Regarding the just right model, authors suggest that Bcl9/9l KO decreases intratumor beta-catenin signaling, which favors tumor formation in the proximal small intestine and shifts the distribution of tumors between the proximal and distal small intestine. Authors' recent Nature Communication paper showed that Porcupine inhibitor also increases proximal tumor formation and shifts the distribution of tumors between the proximal and distal small intestine. Do authors think porcupine inhibitor also decreases beta-catenin signaling in APC deficient tumors? Is there any evidence for this? Porcupine inhibitor does not inhibit the growth of Apc deficient cells.

Some typos in the manuscript:

Line 105: "BCL/9l" should be "BCL9/9l"

Line 179: "does" should be "does not"

Line 264: "Apc fl" should be "Apc fl/+"

Line 316: "Fig. SC&D" should be "Fig. S7C&D"

Line 334: "Fig. S7A" should be "Fig. S8A"

Line 338: "Fig. S7B" should be "Fig. S8B"

Line 445: "sub-optimal" should be "sub-maximal"

Reviewers' comments:

Reviewer #1 (Remarks to the Author):

In the study "Loss of BCL9/9L strongly suppresses Wnt driven tumourigenesis and functionally supports the "just right" Wnt signalling hypothesis" by Prof Sansom and colleagues, the role of BCL9/9L in intestinal homeostasis and regeneration, and tumor formation in the intestine and liver is investigated. It is unequivocally demonstrated that BCL9/9L is required for maintenance of Wnt signaling levels required to sustain Lgr5+ ISCs and drive intestinal regeneration. Furthermore, interesting differences are observed regarding the effect of Bcl9/9l deletion on tumor formation following Apc-loss in the various segments of the bowel. These observations are intriguing but too much emphasis is put on the relationship with Wnt activity levels and various Apc mutations, and just-right signaling. To conclude, the effect of BCL9/9L loss is investigated in beta-catenin driven tumorigenesis in the liver. Altogether the work provides an important advance in the field and is well written albeit with some minor textual inaccuracies. I support publication if some aspects of the work have been adapted.

Concerns:

1. The results of the proximity ligation assay Beta-cat E-cadherin I did not find particularly convincing. Figure 1B should be improved as I do not see a difference when the quantification indicates it should be there.

We thank the reviewer for this observation and agree that the image does not adequately represent the quantified differences observed between WT and *VillinCre^{ER} Bcl9^{fl/fl} Bcl9l^{fl/fl}* intestinal epithelium. We have provided a more representative image which highlights the increased E-cadherin - β -catenin association found in the latter condition.

2. The clonal analysis presented in Fig 2A-C is an interesting highlight of the work and indicates an important impact on stem cell population dynamics of Bcl9/9l. This analysis would be strengthened in case the overlap in allele floxing of *tdtom*, *Bcl9* and *Bcl9L* was directly demonstrated. Furthermore, is there a reduced number of clones detected in Bcl9/9l --/-- condition?

We thank the reviewer for this comment.

In relation to these points, we would note that we have demonstrated that there is a significant reduction in the number of *TdTomato* positive clones per field of view at 21 days post-induction following deletion of BCL9/9L (Figure 2e). Unfortunately, direct assessment of expression of *Bcl9* and *Bcl9l* in histological specimens of normal intestinal tissue through RNAscope was challenging due to the relatively low level of expression. For this reason, we were unable to directly assess the relationship between *TdTomato*, *Bcl9* and *Bcl9l* in serial sections. However, given that *Lgr5* expression was found to be dramatically reduced following BCL9/9L deletion, we have used this as a surrogate

marker for BCL9/9I expression. Using this approach we assessed the expression of *Lgr5* in fully fixed *TdTomato* positive crypts in serial sections taken from *Lgr5Cre^{ER} tdTom⁺ Bcl9^{fl/fl} Bcl9I^{fl/fl}* mice. Of the fully *TdTomato* positive crypts scored across 3 independent mice, 31 out of 35 retained *Lgr5* expression, suggesting that BCL9/9I were not lost from these crypts, we have provided a representative image below. For this reason, we feel that this assay system may in fact underestimate the clonal disadvantage of *Bcl9/9I* null ISCs. This is highlighted in the text.

Reviewer figure 1

Reviewer figure 1: RFP and *Lgr5*-RNAscope staining of serial small intestinal sections from tamoxifen induced *Lgr5Cre^{ER} tdTom^{fl/+} Bcl9^{fl/fl} Bcl9I^{fl/fl}* mice sampled 21 days post-induction. Red arrows indicate crypts that are fully RFP (tomato) positive and also *Lgr5*-positive. The blue arrows point to a crypt that is fully FRP positive but has lost *Lgr5* expression. Scale bar = 50 μ m.

3. The finding that Bcl9/9I loss increases adenoma formation in the proximal small intestine and reduces it in the distal small intestine is very interesting. Similarly, the accompanying manuscript discusses an noteworthy observation in which different *Apc* truncating mutations present with different phenotypes in presence or absence of Bcl9. Yet the interpretation of the authors that this is related to 'just right' signaling and basal levels of Wnt activity at the various compartments of the intestine, and different effects of specific mutations, is simply not indisputably proven. There is lots of hand waving in the text to make this point, but evidently the proper (admittedly very complex) experiments to address this hypothesis have no been performed. The current argument hinges on circumstantial evidence and comparison of multiple model systems, and various segments of the intestine that differ in many more aspects than only basal Wnt levels (or specific *Apc* truncating mutation). For example it might be related to different dependencies of niche factors, different apoptosis thresholds, etc. I propose that the authors clearly indicate that the current interpretation is as speculative as any other possible explanation at this stage and adapt the text, title and abstract to reflect this.

We thank the reviewer for this comment and agree that the 'Just Right' hypothesis of Wnt signalling in CRC is not definitively proven by the data presented here. We have revised the title of the manuscript, along with the abstract and discussion to emphasise that while the data in principle supports the 'Just Right' hypothesis, it remains correlative.

Moreover, to reinforce experimental evidence of the requirement for BCL9/9I in colonic tumour formation, we undertook additional experiments involving localised induction of recombination in *VillinCre^{ER} Apc^{fl/fl}* and *VillinCre^{ER} Apc^{fl/fl} Bcl9^{fl/fl} Bcl9l^{fl/fl}* mice within the colonic epithelium through endoscopic submucosal injection of 4-OH tamoxifen. As was demonstrated in the initial submission (Fig. 5g-h), deletion of BCL9/9I delayed colonic tumour growth. These observations were recapitulated in subsequent experiments, where we were further able to demonstrate that they translate to an extension in survival (Supp fig 6a). Remarkably, those tumours which arise in *VillinCre^{ER} Apc^{fl/fl} Bcl9^{fl/fl} Bcl9l^{fl/fl}* mice retained expression of both *Bcl9* and *Bcl9l*, as determined via RNAScope (Supp fig 6b) – in the context of APC deletion, BCL9/9I are strongly upregulated and readily detectable by RNAScope. These observations would argue that in the absence of BCL9/9I, colonic tumours fail to form, which in turn could be a result of reduced intratumoural Wnt signalling combined with low basal Wnt signalling in the colonic epithelium. Nonetheless, we concede that other factors may play a role, and that the current interpretation of this data with regards to the 'Just Right' hypothesis remains speculative.

4. I wonder if with the basescope assay for exon 14 of *Apc* LOH can be proven. Why can't other loss events be more common?

We thank the reviewer for this comment. There is a significant body of evidence demonstrating that in the case of *Apc* heterozygous mice, the most common mechanism driving loss of the second copy of *Apc* is LOH. By way of example, it has been shown in numerous mouse models of intestinal tumorigenesis harbouring truncating mutations of *Apc*, including *Apc^{Min/+}*, *Apc^{1638/+}* and *Apc^{Δ716/+}*, that in the vast majority of cases tumours lose the second copy of *Apc* by LOH (Haigis et al., 2002; Oshima et al., 1995; Smits et al., 1997).

For this reason we hypothesised that LOH was the most likely mechanism by which the second copy of *Apc* would be lost in our study. To determine whether this is indeed the case, we have quantified the number of tumours from *VillinCre^{ER} Apc^{fl/+}* and *VillinCre^{ER} Apc^{fl/+} Bcl9^{fl/fl} Bcl9l^{fl/fl}* mice which stained positively or negatively for a BaseScope probe targeting *Apc* exon 14. As demonstrated in the figure below the majority of tumours arising in mice of both genotypes were negative for *Apc* – exon 14.

As a proof of principle, and to validate our *Apc* – exon 14 targeting BaseScope probe, we went on to stain intestinal tumours from *VillinCre^{ER} Apc^{fl/+}* and *VillinCre^{ER} Apc^{fl/+} Huwe1^{fl/fl}* mice, having previously demonstrated by PCR that tumours from these mice lose the second copy of *Apc* via LOH (Myant et al., 2017). The figure below also demonstrates that these tumours were negative for *Apc* – exon 14, validating the data generated with our BaseScope probe. Had LOH not been observed in this model system,

whereby the majority of tumours retained expression of *Apc* – exon 14, we would have sought to interrogate other mechanisms by which loss of the second copy of *Apc* could be achieved, such as the mismatch repair mutations as reported in *Apc*^{Min/+} mice that lack *Mlh1* (Shoemaker et al., 2000).

Reviewer figure 2

Reviewer figure 2: a. Staining of small intestinal tumours from *VillinCre^{ER} Apc^{fl/+}* and *VillinCre^{ER} Apc^{fl/+} Bcl9^{fl/fl} Bcl9l^{fl/fl}* mice for *Apc*-exon 14, T denotes tumour and N denotes normal epithelium. Upper panel indicates tumours negative for *Apc*- exon 14 and lower panel indicates tumours positive for *Apc* – exon 14, scale bar = 50µm. b. Quantification of positive and negative stained tumours described in b. n=4 per group, one Way Mann-Whitney *U* test. c. Staining of small intestinal tumours from *VillinCre^{ER} Apc^{fl/+}* and *VillinCre^{ER} Apc^{fl/+} Huwe1^{fl/fl}* mice for *Apc*-exon14, T denotes tumour and N denotes normal epithelium, scale bar = 50µm.

5. Few grammatical errors and typos are present in the text, e.g.:
line 79: 'The beta-catenin-TCF interface is a large and dynamic with...'
line 105: ' BCL/9l' etc.

We thank the reviewer for pointing out these grammatical errors and spelling mistakes in the text, we have corrected them.

Reviewer #2 (Remarks to the Author):

In this comprehensive in vivo study, Gay et al examined the function of Bcl9 and Bcl9l, key components Wnt enhanceosome, in intestinal stem cell fitness, intestinal regeneration, and tumorigenesis. Using conditional Bcl9/9l KO mice, authors showed that BCL9/9l are required for the fitness of intestinal stem cells, for intestinal regeneration, but not for normal tissue homeostasis of the intestine. Loss of Bcl9/9l suppressed APC loss-induced crypt progenitor phenotype with strong reduction of beta-catenin target gene expression. Authors further showed that loss of Bcl9/9l strongly suppressed mutant beta-catenin-driven intestinal transformation and hepatocyte transformation. Authors also made an interesting observation that Bcl9/9l knockout promoted proximal tumor initiation, which functionally supports the just right hypothesis.

Targeting beta-catenin signaling in cancer is complicated by the function of beta-catenin in normal tissue homeostasis, and it is not clear how to generate a therapeutic window. Authors have generated a compelling set of data suggesting that Bcl9/9l are required for high beta-catenin signaling involved in Lgr5+ ISC function and cell transformation, but not for low beta-catenin signaling involved in normal intestinal tissue homeostasis. The study provides strong genetic validation of targeting Bcl9/9l in cancers with APC or beta-catenin mutation. This is especially interesting considering recent findings that Lgr5+ cells might function as tumor stem cells in intestinal tumors. Although the impact of Bcl9 inhibition on other tox tissues such as the bone is still not clear and the study has not touched the function of Bcl9 in tumor maintenance, this work represents a significant step forward and is of importance for the Wnt field. Experiments are beautifully done and data are convincing. It certainly merits publication in Nature Communication.

Regarding the just right model, authors suggest that Bcl9/9l KO decreases intratumor

beta-catenin signaling, which favors tumor formation in the proximal small intestine and shifts the distribution of tumors between the proximal and distal small intestine. Authors' recent Nature Communication paper showed that Porcupine inhibitor also increases proximal tumor formation and shifts the distribution of tumors between the proximal and distal small intestine. Do authors think porcupine inhibitor also decreases beta-catenin signaling in APC deficient tumors? Is there any evidence for this? Porcupine inhibitor does not inhibit the growth of Apc deficient cells.

We thank the reviewer for this comment regarding our previous work describing the impact of porcupine inhibition upon intestinal tumorigenesis. In order to address whether porcupine inhibition decreases β -catenin signalling in APC deficient tumours, we have stained tumours from *Lgr5Cre^{ER} Apc^{fl/fl}* and intestinal tissue from *VillinCre^{ER} Apc^{fl/fl}* mice treated with porcupine inhibitor or the appropriate vehicle control for a number of Wnt target genes, including *Lgr5*, *Axin2* and CD44, as well as for nuclear accumulation of β -catenin itself. Using this approach, we were able to demonstrate that there is no discernible difference in the expression of these Wnt target genes or the localisation of β -catenin in intestinal epithelium of these mice upon porcupine inhibition. This is detailed in the panel below.

Moreover, as highlighted by the reviewer, we have previously shown that inhibition of porcupine does not affect proliferation in Apc deficient cells (Huels et al., 2018).

Reviewer figure 3

Reviewer figure 3: β -catenin, Axin2-, Lgr5-RNAscope and CD44 staining from a. small intestinal sections of *VillinCre^{ER} Apc^{fl/fl}* mice treated with 5mg/kg twice daily with LGK974 or vehicle from 24 hours post-induction and sampled 4-days later. b. Small intestinal

tumours from *Lgr5Cre^{ER} Apc^{fl/fl}* mice treated with 5mg/kg LGK974 twice daily from 24 hours post-induction until clinical end-point or untreated controls. Scale bar = 50µm.

Some typos in the manuscript:

Line 105: "BCL/9I" should be "BCL9/9I"

Line 179: "does" should be "does not"

Line 264: "Apc fl" should be "Apc fl/+"

Line 316: "Fig. SC&D" should be "Fig. S7C&D"

Line 334: "Fig. S7A" should be "Fig. S8A"

Line 338: "Fig. S7B" should be "Fig. S8B"

Line 445: "sub-optimal" should be "sub-maximal"

We thank the reviewer for pointing out these grammatical errors and spelling mistakes in the text, we have corrected them.

References:

Haigis, K. M., Caya, J. G., Reichelderfer, M., & Dove, W. F. (2002). Intestinal adenomas can develop with a stable karyotype and stable microsatellites. *Proceedings of the National Academy of Sciences of the United States of America*, *99*(13), 8927–31. <https://doi.org/10.1073/pnas.132275099>

Huels, D. J., Bruens, L., Hodder, M. C., Cammareri, P., Campbell, A. D., Ridgway, R. A., ... Sansom, O. J. (2018). Wnt ligands influence tumour initiation by controlling the number of intestinal stem cells. *Nature Communications*, *9*(1), 1132. <https://doi.org/10.1038/s41467-018-03426-2>

Myant, K. B., Cammareri, P., Hodder, M. C., Wills, J., Kriegsheim, A. Von, Rashid, M., ... Sansom, O. J. (2017). HUWE 1 is a critical colonic tumour suppressor gene that prevents MYC signalling, DNA damage accumulation and tumour initiation. *EMBO Molecular Medicine*, *9*(2), 181–197. <https://doi.org/10.15252/emmm.201606684>

Oshima, M., Oshima, H., Kitagawa, K., Kobayashi, M., Itakura, C., & Taketo, M. (1995). Loss of Apc heterozygosity and abnormal tissue building in nascent intestinal polyps in mice carrying a truncated Apc gene. *Proceedings of the National Academy of Sciences*, *92*(10), 4482–4486. <https://doi.org/10.1073/pnas.92.10.4482>

Shoemaker, A. R., Haigis, K. M., Baker, S. M., Dudley, S., Liskay, R. M., & Dove, W. F. (2000). Mlh1 deficiency enhances several phenotypes of Apc(Min/+) mice. *Oncogene*, *19*(23), 2774–2779. <https://doi.org/10.1038/sj.onc.1203574>

Smits, R., Kartheuser, A., Jagmohan-Changur, S., Leblanc, V., Breukel, C., De Vries, A., ... Fodde, R. (1997). Loss of Apc and the entire chromosome 18 but absence of mutations at the Ras and Tp53 genes in intestinal tumors from Apc1638N, a mouse model for Apc-driven carcinogenesis. *Carcinogenesis*, *18*(2), 321–327. <https://doi.org/10.1093/carcin/18.2.321>

REVIEWERS' COMMENTS:

Reviewer #1 (Remarks to the Author):

All my concerns have been successfully addressed.

Reviewer #2 (Remarks to the Author):

I am fine with the revised manuscript.

Authors partially answered my question. Porcupine inhibitor does not affect Wnt signaling in APC deficient cells, which is probably expected. On the other hand, if porcupine inhibitor does not affect Wnt signaling in APC deficient cells, how can it shift the distribution of tumors between the proximal and distal small intestine? I guess authors do not have a good answer for this. On the other hand, this is really a question directly related to authors' previous publication. It highlights the complexity of the system, so I am fine to leave it as it is.